# Egocentric Video-Language Pretraining

**Kevin Qinghong Lin**[1], **Alex Jinpeng Wang**[1], **Mattia Soldan**[3], **Michael Wray**[2],
**Rui Yan**[1], **Eric Zhongcong Xu**[1], **Difei Gao**[1], **Rongcheng Tu**[4],
**Wenzhe Zhao**[4], **Weijie Kong**[4], **Chengfei Cai**[4], **Hongfa Wang**[4],
**Dima Damen**[2], **Bernard Ghanem**[3], **Wei Liu**[4], and **Mike Zheng Shou**[1✉]

[1]Show Lab, National University of Singapore    [2]University of Bristol
[3]King Abdullah University of Science and Technology    [4]Tencent Data Platform

## Abstract

Video-Language Pretraining (VLP), which aims to learn transferable representation to advance a wide range of video-text downstream tasks, has recently received increasing attention. Best performing works rely on large-scale, 3rd-person video-text datasets, such as HowTo100M. In this work, we exploit the recently released Ego4D dataset to pioneer Egocentric VLP along three directions. (i) We create EgoClip, a 1st-person video-text pretraining dataset comprising 3.8M clip-text pairs well-chosen from Ego4D, covering a large variety of human daily activities. (ii) We propose a novel pretraining objective, dubbed EgoNCE, which adapts video-text contrastive learning to the egocentric domain by mining egocentric-aware positive and negative samples. (iii) We introduce EgoMCQ, a development benchmark that is close to EgoClip and hence can support effective validation and fast exploration of our design decisions in EgoClip and EgoNCE. Furthermore, we demonstrate strong performance on five egocentric downstream tasks across three datasets: video-text retrieval on EPIC-KITCHENS-100; action recognition on Charades-Ego; natural language query, moment query, and object state change classification on Ego4D challenge benchmarks. The dataset and code are available at https://github.com/showlab/EgoVLP.

## 1 Introduction

With the recent interest boom in computer vision and natural language processing, Video-Language Pretraining (VLP) has prevailed, which aims to learn strong and transferable video-language representation for powering a broad spectrum of video-text downstream tasks, such as video-text retrieval [1, 2, 3], video question answering [4, 5, 6], and video captioning [7, 8, 9]. The success of VLP mainly stems from the availability of large-scale open-world video-text datasets [10], which subsume a large number of videos sourced from the Web (e.g., YouTube) and pair videos with associated textual information. For instance, HowTo100M [10] collects 134K hours of instructional videos accompanied by noisy narrations yielded from Automatic Speech Recognition (ASR). WebVid-2M [3] scrapes 2.5M descriptive videos with well-formed long captions.

Despite reaching an impressive data scale, videos in those existing video-text pretraining datasets are often of 3rd-person views and may have been edited before posting on the Web. Yet, there is a noticeable domain gap between the existing video-text pretraining datasets and 1st-person view videos such as those videos captured by wearable cameras or smart glasses. Egocentric video has received increasing interests from the academia (e.g., activity recognition [11], activity anticipation [12], and video summarization [13]) and industry (various applications in robotics and augmented reality).

---

✉: Corresponding Author.

36th Conference on Neural Information Processing Systems (NeurIPS 2022).

| Dataset | Ego? | Domain | Dur (hrs) | # Clips | # Texts | Example |
|---|---|---|---|---|---|---|
| MSR-VTT [1] | ✗ | diverse | 40 | 10K | 200K | |
| YouCook2 [16] | ✗ | cooking | 176 | 14K | 14K | |
| ActivityNet Captions [7] | ✗ | action | 849 | 100K | 100K | |
| WebVid-2M [3] | ✗ | diverse | 13K | 2.5M | 2.5M | |
| HowTo100M [10] | ✗ | instructional | 134K | 136M | 136M | 3rd-person view |
| Charades-Ego [17] | ✓ | home | 34 | 30K | 30K | |
| UT-Ego [18] | ✓ | diverse | 37 | 11K | 11K | |
| Disneyworld [19] | ✓ | disneyland | 42 | 15K | 15K | |
| EPIC-KITCHENS-100 [20] | ✓ | kitchen | 100 | 90K | 90K | |
| **EgoClip** | ✓ | **diverse** | **2.9K** | **3.8M** | **3.8M** | 1st-person view |

Table 1: Comparison of our proposed EgoClip pretraining dataset against the mainstream video-language datasets (top) and egocentric datasets (bottom).

However, due to such a domain gap, directly transferring the existing VLP models to egocentric downstream tasks cannot fully unleash the potential of large-scale pretraining approaches, which we have confirmed in the later experimental section. To bridge this gap, we are motivated to develop Egocentric VLP models, which can greatly benefit various egocentric video downstream applications.

However, existing egocentric video datasets are of small scale and domain-specific, making Egocentric VLP prohibitive. As illustrated in Tab. 1, the formerly largest egocentric video dataset EPIC-KITCHENS-100 [14] focuses on kitchens scenarios and its size is far smaller than those of the 3rd-person pretraining sets WebVid-2M [3] and HowTo100M [10]. Fortunately, with the recent introduction of the massive-scale egocentric video dataset Ego4D [15], it becomes possible to unlock Egocentric VLP. Ego4D consists of 3,670 hours of videos with manually annotated narrations from 74 worldwide locations, covering a large variety of daily-life scenarios and activities.

In this work, roused by the favorable scale and diversity of Ego4D, we make a significant effort to pave the way for Egocentric VLP with the following steps:
**(i)** To address the aforementioned issue of lacking a suitable large-scale egocentric video-language pretraining dataset, we create a video-text pretraining dataset **EgoClip** which contains a total of 3.8M clean 1st-person clip-text pairs selected from Ego4D and covers diverse human daily activities.
**(ii)** To make full use of EgoClip for video-text representation learning, we propose a novel video-text contrastive objective **EgoNCE** to address unique challenges in egocentric pretraining datasets.
**(iii)** We create a development benchmark i.e., Egocentric Multiple-Choices-Question, dubbed **EgoMCQ**, which contains 39K questions created from Ego4D and focuses on evaluating video-text alignment. In contrast to other downstream benchmarks, EgoMCQ has a less discrepancy from EgoClip, powering us to accurately validate and quickly iterate our designs of EgoClip and EgoNCE.
**(iv)** We conduct extensive experiments to demonstrate the superiority of Egocentric VLP by transferring our pretrained representation to five egocentric downstream benchmarks and achieving state-of-the-art performance: $59.4\%$ nDCG on video-text retrieval of EPIC-KITCHENS-100 [14] [1], $32.1\%$ mAP on action recognition of Charades-Ego [17], and significant boosts over three Ego4D challenges [2]: natural language query, moment query and object state change classification.

## 2  Related Work

**Video-Language Pretraining.** The introduction of large-scale video-text datasets [10, 3] has enabled the emergence of VLP approaches to improve the video-text representation for various vision-language tasks [21, 22, 4], such as MIL-NCE which [23] proposes to match clips with multiple captions close in temporal to adapt the video-text misalignment of HowTo100M [10]. Dominant VLP methods can be classified into two groups, namely: joint- and dual-encoders. The former combines videos and texts as a single input to the encoder that performs the multimodal fusion. For instance, [24, 25] concatenate videos and texts together before feeding them to a unified transformer. Conversely, methods like [3, 26] exploit dual encoders to independently project the video and text inputs into a common space and minimize the distance between the paired representations. These approaches are preferred in retrieval settings as they allow for efficient indexing of a single

---

[1]Egocentric VLP won championship on Multi-Instance Retrieval, EPIC-Kitchens Challenges @ CVPR 2022.
[2]Egocentric VLP won championship on OSCC and 2nd place on NLQ, Ego4D Challenges @ CVPR 2022.

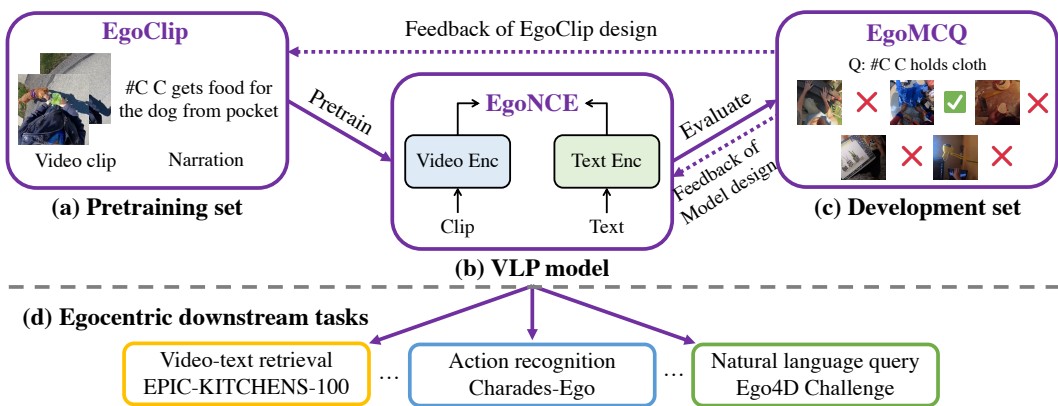

Figure 1: Our Egocentric VLP includes: (a) the pretraining set EgoClip, (b) the VLP model, and (c) the development set EgoMCQ. We use EgoClip to pretrain a VLP model with the EgoNCE loss and then evaluate on EgoMCQ. According to the feedback, we iteratively refine our designs of (a) and (b). We then transfer the pretrained model to downstream tasks relevant to the egocentric domain.

modality [27, 28]. For example, Frozen [3] employs two separate transformers to encode video and text features and aligns them by video-text InfoNCE [29]. In our work, we adopt the Frozen [3] but extend its InfoNCE to EgoNCE via positive and negative sampling for egocentric-friendly pretraining.

**Egocentric Video Datasets.** Egocentric videos, collected by participants using wearable cameras, offer a natural perspective of people's daily activities and raise a range of challenging research topics [11, 12, 30]. Several egocentric video datasets have been developed in decades, e.g., [20, 17, 31]. However, since the collection of egocentric videos is expensive, previous egocentric datasets tend to be small-scale and domain-specific. These limitations hinder 1st-person view research and fail to match the progress of 3rd-person counterparts, such as VLP [23, 24, 3]. Recently, a massive egocentric video dataset Ego4D [15] has been released, which consists of $3,670$ hours of videos collected by 931 people from 74 worldwide locations in 9 different countries, where most videos are accompanied by narrations, audio, 3D meshes, and more. Furthermore, Ego4D introduces a suite of new challenging benchmarks (e.g., Natural language query and moment query) to fully explore the 1st-person visual experience. With this step-changing dataset and benchmarks, Ego4D would lead to a new research surge on egocentric visual perception.

## 3 EgoClip: An Egocentric Video-Language Pretraining Dataset

**Data curation.** For our EgoClip dataset, we source data from Ego4D [15], which contains $9,645$ untrimmed videos of varying lengths from 5 sec to 7 hrs. From these videos, most are associated with *dense timestamp-level narrations* assigned by two different annotators, describing the camera wearer's activities and interactions with objects. For example, the narration "#C C puts the scrapper down." corresponds to video content that occurred at $3.70s$, where "#C" refers to the camera-wearer. Notably, narrations in Ego4D are well-aligned with the videos, both temporally and visually. Prior pretraining datasets are characterized by a much greater level of temporal misalignment between the video and text (e.g., HowTo100M [10] narrations are scraped from ASR, yielding sentences misaligned or even unrelated to video content). We first filter Ego4D videos with missing narrations ($7.4\%$ of the total video duration) and exclude videos that belong to the validation and test sets of the Ego4D benchmark challenge [15] (a further $23.9\%$ of the total video duration). Next, we retain textual annotation from both narrators in EgoClip, allowing us to consider narration diversity when pairing video and text for pretraining purposes. Finally, we adopt several criteria to filter the video and textual narrations, further reducing noise (detailed steps are provided in Supplementary B.1). Overall, this procedure yields 2.9K hours of videos with 3.85 million narrations which cover 2927 hours of video from 129 different scenarios. EgoClip has 21.9 clips per minute with an average clip length of $1.0$ seconds and a standard deviation of $0.9$ seconds (the longest clip is up to 60s). Additional analyses are included in the Supplementary B.3.

**Creation of clip-text pairs.** Clip-text pairs are the common data format for VLP, but are usually not present in untrimmed video datasets with only a weak matching between narrations captions and videos. This was first discussed in HowTo100M [10], which pairs subtitles to video clips with

corresponding time intervals to produce noisy pairs. This is not suitable for Ego4D since each narration is annotated with a single timestamp rather than an interval. Thus, we design *a contextual variable-length clip pairing strategy*. Formally, narrations per video in Ego4D are organized as a sequence of sentences $\{\mathcal{T}_0, \cdots, \mathcal{T}_n\}$ with exact timestamps $\{t_0, \cdots, t_n\}$, indicating an event $i$ described by $\mathcal{T}_i$ happened in the moment $t_i$. For a narration $\mathcal{T}_i$ with timestamp $t_i$, we pair a clip $\mathcal{V}_i$ with following start and end timepoints:

$$[t_i^{start}, t_i^{end}] = [t_i - \beta_i/2\alpha, \ t_i + \beta_i/2\alpha], \tag{1}$$

which represents a window centered around the timestamp $t_i$ with temporal duration equal to $\beta_i/\alpha$. $\beta_i$ is an adjustable parameter equal to the average temporal distance between pairs of consecutive narrations, i.e., $\sum_{j=0}^{n-1}(t_{j+1} - t_j)/n$. We compute $\beta_i$ on a per video basis. Conversely, $\alpha$ is a scale factor computed as the average of all $\beta_i$ across all videos in the EgoClip ($\alpha = 4.9$ seconds). Intuitively, Eq. 1 is derived from three observations: **(i)** Centering $t_i$ helps involve prior information about the event $i$; **(ii)** $\beta_i$ measures the clip duration according to its scenario, such as longer clips watching television (352.9 seconds) v.s. shorter clips harvesting crops (0.9 seconds); **(iii)** $\alpha$ controls the context granularity of clips (e.g., a large $\alpha$ pays more attention to rapid, atomic actions). We ablate these design choices in our experimental section.

## 4 Video-Language Pretraining Model

To efficiently transfer video-language representation to egocentric downstream tasks (e.g., video-text retrieval on EPIC-KITCHENS-100 [20]), We prefer the dual-encoder (discussed in Sec. 2) as our VLP model architecture. In particular, we emphasize devising a general pretraining objective EgoNCE to adapt the existing VLP model to the egocentric domain (e.g., EgoClip).

### 4.1 Architecture: Dual-encoder Pipeline

We choose Frozen [3] as our pretraining architecture. Frozen [3] design encompasses an elegant and simple dual encoder strategy (one per modality) which has favorable characteristics (e.g., indexability and efficiency [27, 28]). Note that this allows us to use our pretrained network in single-modality tasks (e.g., video-only tasks). In practice, the video encoder adopts the TimeSformer [32] architecture, while the text encoder builds upon DistillBERT [33]. However, our approach is not limited to the encoder's design (e.g., the video backbone can be replaced by SlowFast [34] or Video Swin [35]). In the rest of the paper we adopt this notation: $(\mathcal{V}_i, \mathcal{T}_i)$ represents the video-text input to the model, while $\mathbf{v}_i$ and $\mathbf{t}_i$ are used to identify the video and text embeddings.

### 4.2 EgoNCE: An Egocentric-friendly Pretraining Objective

A common pretraining objective for the dual-encoder VLP is **InfoNCE** [29], where the matching visual-text pairs in the batch are treated as positives while all other pairwise combinations in the batch are regarded as negatives. Formally, within a batch $\mathcal{B} = \{1, \cdots, N\}$, InfoNCE is computed by the sum of the video-to-text loss $\mathcal{L}_{v2t}$ and text-to-video loss $\mathcal{L}_{t2v}$. For simplicity, we only formulate $\mathcal{L}_{v2t}$, whereas $\mathcal{L}_{t2v}$ is defined in a symmetric way:

$$\mathcal{L}_{v2t} = -\frac{1}{|\mathcal{B}|} \sum_{i \in \mathcal{B}} \log \frac{\exp(\mathbf{v}_i^T \mathbf{t}_i/\tau)}{\sum_{j \in \mathcal{B}} \exp(\mathbf{v}_i^T \mathbf{t}_j/\tau)}, \tag{2}$$

where the $i$-th video embedding $\mathbf{v}_i$ and $j$-th text embedding $\mathbf{t}_j$ are $L_2$ normalized features, and $\tau$ is a temperature factor.

However, this simple objective performs not well on large-scale video-text datasets like HowTo100M [10] due to the serious misalignment between the two modalities of data. Therefore, [36] proposes MIL-NCE which treats temporal nearest captions as positive samples.

In this work, our 1st-person human daily activity dataset, i.e. EgoClip, presents two unique challenges compared to the existing 3rd-person view video-text datasets: **Challenge (i)**: The same action often occurs in different scenarios (e.g., "unlock the phone" could happen when "lying in bed" or "walking outdoors"). **Challenge (ii)**: Often, different actions appearing in the same scenario tend to have indistinguishable visual differences (e.g., when "working in front of the laptop", "typing on the keyboard" or "moving the mouse" have similar feature representations).

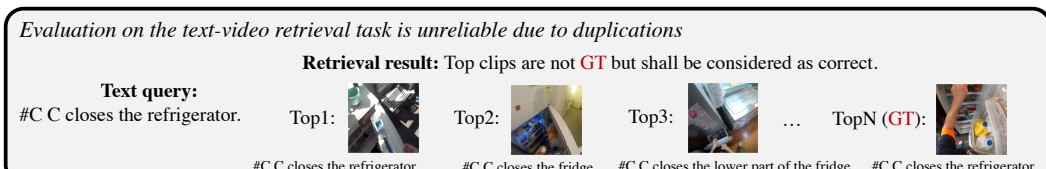

| EgoMCQ | Inter-video | | | | | Intra-video | | | | |
|---|---|---|---|---|---|---|---|---|---|---|
| **Text query** | #C C picks the silicone sealant | | | | | #C C carries paint bucket down the ladder | | | | |
| **Select the correct video clip from 5 candidates** | (a) | (b) | (c) | (d) | (e) | (a) | (b) | (c) | (d) | (e) |
| **Answer with GT** | #C C places the camping seat down | #C C holds the power drill with both hands. | #C C picks the silicone sealant | #C C takes a stone | #C C cuts the green bean into pieces | #C C holds paintbrush with both hands | #C C turns paintbrush in his left hand | #C C shifts paintbrush to right hand | #C C drops paintbrush on paint bucket | #C C carries paint bucket down the ladder |
| | ✗ | ✗ | ✅ | ✗ | ✗ | ✗ | ✗ | ✗ | ✗ | ✅ |

Figure 2: Design of the Egocentric VLP development set. **Top:** An illustration of why the task of text-video retrieval is not suitable; **Bottom:** Two settings of EgoMCQ. **Left-bottom:** The "inter-video" setting, each question contains 5 clips from different videos. **Right-bottom:** The "intra-video" setting, each question contains 5 contiguous clips from the same video, making it more challenging.

To overcome these two unique challenges, we propose a novel EgoNCE training objective which takes into account two simple yet efficient sampling strategies based on the vanilla InfoNCE.

**Action-aware Positive Sampling.** In this work, we make a reasonable assumption that the critical elements in linking visual actions to textual narrations are verbs and objects mentioned in the narrations (e.g., "drinking coffee" and "opening fridge"). Following this assumption, we can devise a clever method to address challenge (i). Specifically, for each narration, we identify its nouns and verbs and merge synonym words based on the Ego4D taxonomy dictionary [15], a thesaurus recording meaningful nouns/verbs in Ego4D narrations. Then, batch samples that shared at least one noun and at least one verb are treated as positive samples. At last, for the sample $i$, we define its positive samples set within batch $\mathcal{B}$ as $\mathcal{P}_i = \{j \in \mathcal{B} \mid \text{noun}(j) \cap \text{noun}(i) \neq \varnothing, \text{verb}(j) \cap \text{verb}(i) \neq \varnothing\}$.

**Scene-aware Negative Sampling.** To address challenge (ii), we consider different actions in the same scenario as hard negative samples. Specifically, for each video clip $i$, we sample an adjacent clip $i' \in \mathcal{N}(i)$, which is close to $i$ in time within the same video. We augment the original batch $\mathcal{B}$ with such hard negative samples and each sample $i$ in $\mathcal{B}$ has its negative counterparts $i'$. Hence the batch is updated as $\widetilde{\mathcal{B}} = \{\underbrace{1, 2, \cdots N}_{\mathcal{B}}, \underbrace{1', 2', \cdots, N'}_{\mathcal{N}(\mathcal{B})}\}$.

With these two sampling strategies, our new pretraining objective **EgoNCE** can be formulated as:

$$\mathcal{L}_{\text{v2t}}^{\text{ego}} = -\frac{1}{|\widetilde{\mathcal{B}}|} \sum_{i \in \widetilde{\mathcal{B}}} \log \frac{\sum_{k \in \mathcal{P}_i} \exp(\mathbf{v}_i^T \mathbf{t}_k / \tau)}{\sum_{j \in \mathcal{B}} \left( \exp(\mathbf{v}_i^T \mathbf{t}_j / \tau) + \exp(\mathbf{v}_i^T \mathbf{t}_{j'} / \tau) \right)}. \tag{3}$$

Here the item in purple corresponds to our proposed action-aware positive samples and blue corresponds to our proposed scene-aware negative samples. EgoNCE provides a general extension to adapt the existing VLP models for video-text pretraining datasets in the egocentric domain.

## 5  EgoMCQ: A Benchmark for Egocentric VLP Development

**The need for a development benchmark.** We find that most egocentric benchmarks are domain-specific and focus on single-modality tasks (see Tab. 1). However, our purpose is to exploit Ego4D's diversity to learn rich video-text representations. Hence, to validate our design choices of the pretraining dataset (e.g., EgoClip), and model (e.g., EgoNCE), it is essential to measure performance on a benchmark highly aligned with the pretraining task. Therefore, we propose EgoMCQ, a new egocentric benchmark for reliable and fast developments of Egocentric VLP.

**Data source.** We start from the Ego4D data excluded from constructing the EgoClip, which mainly covers the validation set of the Ego4D challenge benchmarks. Additionally, to assure that the scene is not visible during pretraining, we manually remove videos that share multiple views with the videos in EgoClip. To ensure diversity, we randomly select one annotator's narration for each video. We follow the same clip pairing strategy as Eq. 1 to be consistent with the data format of EgoClip.

**Benchmarking task design.** To determine the task for development, we first consider video-text retrieval since it highly aligns with the VLP pretraining objective. However, as depicted in the top half of Fig. 2, for an action (e.g., close the refrigerator), there are substantial duplicates or semantically similar captions in Ego4D. This can cause issues in retrieval evaluation [37] making model training unreliable. A straightforward approach to prevent this is deduplication (dedup), but it is challenging to devise a dedup criterion and perform well in the retrieval settings of a "one-to-whole validation set". Therefore, we select the *Multiple-Choice Questions (MCQ)* task for development since repetitions are highly unlikely given a small number of answers.

**Grouping strategies.** To set up the MCQ task, a naive construction randomly groups five video clips to form options for a question. But we find randomly grouping is not challenging since options are highly likely to come from different videos and vary widely in content. We redefine this basic setting as "**inter-video**" and ensure that the five clips originate from different videos, aiming to distinguish instances from different scenarios (the left-bottom of Fig. 2). Furthermore, we propose a more challenging setting "**intra-video**" by grouping five continuous clips together. This setting is regarded as a specific form of video-text localization focused on fine-grained context clues, such as hand interaction (the right-bottom of Fig. 2). Dedup is performed within five options for each question for reliable assessment (see Supp. C.1) and we adopt accuracy as the EgoMCQ metric.

**Statistics.** We finalize 39K questions covering 198K narrations with 468 hours of video, where the "inter-video" has 24K questions covering 290.3 hours of videos. And the "intra-video" has 15K questions and covers 178.3 hours of videos. The average duration among the five options is 34.2 seconds (More statistics of EgoMCQ are shown in Supplementary C.3).

## 6   Experiments

We assess our Egocentric VLP along two directions: **(i)** We conduct an extensive analysis to explore key components of Egocentric VLP (e.g., EgoClip, EgoNCE, and EgoMCQ); **(ii)** we transfer our pretrained model to various downstream tasks to validate the quality of our video-text representation.

### 6.1   Benchmarks and Settings

We evaluate our VLP model on five egocentric benchmarks, spanning video-text tasks and pure video tasks, across three different datasets. We briefly describe each task below.

**Multi-Instance Retrieval of EPIC-KITCHENS-100.** This task is modelled as a video-text retrieval which considers the semantic overlap between different videos narrations, where multiple videos may correspond to the same narration. The training set contains 67.2K clips and validation set contains 9.7K clips. The evaluation metrics are mean Average Precision (mAP) and the normalized Discounted Cumulative Gain (nDCG).

**Natural Language Query of Ego4D Challenges.** The Natural Language Query task is modelled as a natural language grounding problem [38, 39, 40]. Given a language query and a video, the task aims at localizing the temporal interval within the video, in which the answer is deducible. The training set contains 11.3K queries annotated from 1K clips for this task, while the validation contains 3.9K queries collected from 0.3K clips. The evaluation metric is Recall@$K$ for IoU=$\theta$ (R@$K$-IoU=$\theta$) [38] where $\theta$ is a threshold. We evaluate for $K \in \{1, 5\}$ and $\theta \in \{0.3, 0.5\}$.

**Action Recognition of Charades-Ego.** This dataset has 64K instances, spanning 1st-person and 3rd-person views and covering 157 activity categories for training. We train and evaluate only on the 1st-person videos. The validation set contains 847 videos for classification and each video belongs to multiple classes. The evaluation metric is mAP.

**Moment Query of Ego4D Challenges.** The Moment Query task is a video-only task modelled as Temporal Action Localization [11]. Given a particular high-level activity category, the task solution consists of retrieving all the possible temporal windows where the activity occurs. The training set

| Clip creation strategy | Clip's length (s) Avg ± Std | EgoMCQ Acc (%) Inter-video  Intra-video | Zero-shot T↔V Retrieval [20] mAP (avg)      nDCG (avg) |
|---|---|---|---|
| (a) $[t_i, t_i+\alpha]$ | $5.0 \pm 0.0$ | 87.66   39.72 | 19.6    12.3 |
| (b) $[t_i-\alpha/2, t_i+\alpha/2]$ | $5.0 \pm 0.0$ | 89.23   41.68 | 20.6    13.7 |
| (c) $[t_{i-1}, t_{i+1}]$ | $10.0 \pm 38.2$ | 88.13   40.62 | 20.6    13.7 |
| (d) $[t_i-\beta_i/2, t_i+\beta_i/2]$ | $4.9 \pm 4.7$ | 89.74   44.82 | 21.1    14.5 |
| (e) $[t_i-\beta_i/4, t_i+\beta_i/4]$ | $2.4 \pm 2.4$ | **90.23**   49.67 | 21.9    15.3 |
| (f) $[t_i-\beta_i/2\alpha, t_i+\beta_i/2\alpha]$ | $1.0 \pm 0.9$ | 89.36   **51.51** | **22.1**    **15.5** |

Table 2: Results on our development set EgoMCQ and video-text retrieval on EPIC-KITCHENS-100 when using different strategies in the creation of EgoClip, where $t_i$, $\alpha$, $\beta_i$ are defined in Eq. 1. In all experiments, we bold the **best results** and underlined the second best results.

contains 13.6K instances from 1.5K clips, while the validation set contains 4.3K instances from 0.5K clips. The evaluation metrics are mAP and R@$K$-IoU$=\theta$ for $K \in \{1, 5\}$ and $\theta \in \{0.3, 0.5, 0.7\}$.

**Object State Change Classification (OSCC) of Ego4D Challenges.** This OSCC task is modelled as an (N+1)-way classification aiming to identify an object's state change in a given video. The training and val. sets contain 41K and 28K clips, respectively. The evaluation metric is accuracy.

**Implementation Details.** Our codebase is based on the official Frozen [3] one and retains the same settings unless specified. During pretraining, we sample 4 frames for each clip, and use the Adam optimizer [41] with a learning rate of $3 \times 10^{-5}$. To select the best method we pretrain our architecture for 10 epochs and use the best performing model on the EgoMCQ benchmark. Pretraining takes two days on 32 A100 GPUs ($1,536$ GPU hrs).

## 6.2 Ablation Studies

**Ablation of the strategy used when creating EgoClip.** We validate our proposed strategies, i.e., Eq.1 in Tab. 2, by comparing the following variants: (a) fixed length $\alpha$, start at timestamp; (b) fixed length $\alpha$, center at timestamp; (c) variable clip, start and end by adjacent timestamps; (d) our proposed strategy, scaled by 2; (e) our proposed strategy, scaled by 4; (f) our proposed strategy.

We consider that a good pretraining dataset creation strategy should satisfy: **(1)** the VLP model trained on EgoClip should be able to well distinguish instances in EgoMCQ with the same data format; **(2)** the VLP model pretrained on EgoClip with the specific clip creation strategy should perform well on public downstream tasks (e.g., video-text retrieval on [20] and zero-shot for efficiency).

We draw several conclusions from Tab. 2: **(i)** The performance of EgoMCQ is well aligned with the zero-shot result on EPIC-KITCHENS-100, especially minor gain on downstream but noticeable on EgoMCQ, which means EgoMCQ provides valid feedback and is suitable as a development set. **(ii)** Under the same clip length $\alpha$, (b) surpassing (a) proves that centering at timestamp includes prior information is helpful. **(iii)** Variable-length clips make a big difference, as shown in (c) and (d).

Notably, with our designed $\beta_i$, (d) outperforms (b) with a similar average clip length, which validates our key idea of "contextual varied clip length". **(iv)** Based on (d), (e), and (f), we found a proper scale factor greater than 1 is preferred, which helps focus on a large of instantaneous actions densely labeled by Ego4D [15]. These ablation studies demonstrate the effectiveness of our proposed EgoClip creation strategy and EgoMCQ for development.

| Variants | Accuracy (%) Intra-video    Inter-video | |
|---|---|---|
| InfoNCE | 89.4 | 51.5 |
| (a) w/ Pos, noun | 82.9 (6.5 ↓) | 42.3 (9.2 ↓) |
| (b) w/ Pos, verb | 86.9 (2.5 ↓) | 50.5 (1.0 ↓) |
| (c) w/ Pos, noun & verb | 89.7 (0.4 ↑) | 53.6 (2.1 ↑) |
| (d) w/ Neg, random | 88.3 (1.1 ↓) | 49.9 (1.6 ↓) |
| (e) w/ Neg, within video | 89.7 (0.3 ↑) | 53.0 (1.5 ↑) |
| (f) w/ Neg, within 1 min | 89.5 (0.2 ↑) | 54.5 (3.0 ↑) |
| (g) w/ Pos & Neg, **EgoNCE** | **90.6** (1.3 ↑) | **57.2** (5.7 ↑) |

Table 3: EgoNCE sampling strategy ablation. We evaluate accuracy performance on our development benchmark EgoMCQ.

**Effect of EgoNCE.** In this section, we evaluate the effect of the proposed sampling strategies for the EgoNCE objective (Eq. 3) on EgoMCQ and compare against a vanilla InfoNCE loss (Eq. 2). We ablate several configurations for positive and negative sampling strategies. The sampling strategy for positive pairs exploits language cues, while negative pairs rely on temporal, visual cues. Given a text-video pair, we regard other text-video pairs as positive if the textual narrations: (a) share at

---

[3]https://github.com/m-bain/frozen-in-time

| Methods | Vis Enc Input | # Frames | Vis-text PT | mAP (%) | | | nDCG (%) | | |
|---|---|---|---|---|---|---|---|---|---|
| | | | | V→T | T→V | Avg | V→T | T→V | Avg |
| Random | - | - | - | 5.7 | 5.6 | 5.7 | 10.8 | 10.9 | 10.9. |
| MI-MM | S3D [42] | 32 | HowTo100M | 34.8 | 23.6 | 29.2 | 47.1 | 42.4 | 44.7 |
| MME [43] | TBN † [14] | 25 | - | 43.0 | 34.0 | 38.5 | 50.1 | 46.9 | 48.5 |
| JPoSE [43] | TBN † [14] | 25 | - | 49.9 | 38.1 | 44.0 | 55.5 | 51.6 | 53.5 |
| Frozen | Raw Videos | 4 | - | 38.8 | 29.7 | 34.2 | 50.5 | 48.3 | 49.4 |
| Frozen | Raw Videos | 4 | HowTo100M | 39.2 | 30.1 | 34.7 | 50.7 | 48.7 | 49.7 |
| Frozen | Raw Videos | 4 | CC3M+WebVid-2M | 41.2 | 31.6 | 36.4 | 52.7 | 50.2 | 51.4 |
| Frozen | Raw Videos | 4 | EgoClip | 44.5 | 34.7 | 39.6 | 55.7 | 52.9 | 54.3 |
| Frozen+EgoNCE | Raw Videos | 4 | EgoClip | **45.1** | **35.3** | **40.2** | **56.2** | **53.5** | **54.8** |
| Frozen | Raw Videos | 16 | CC3M+WebVid-2M | 45.8 | 36.0 | 40.9 | 57.2 | 54.3 | 55.8 |
| Frozen+EgoNCE | Raw Videos | 16 | EgoClip | **49.9** | **40.1** | **45.0** | **60.9** | **57.9** | **59.4** |
| Frozen | Raw Videos | 4 | HowTo100M | 6.8 | 6.3 | 6.5 | 11.6 | 12.8 | 12.2 |
| Frozen | Raw Videos | 4 | CC3M+WebVid-2M | 8.6 | 7.4 | 8.0 | 14.5 | 14.6 | 14.5 |
| Frozen | Raw Videos | 4 | EgoClip | 17.9 | 13.1 | 15.5 | 23.0 | 21.2 | 22.1 |
| Frozen+EgoNCE | Raw Videos | 4 | EgoClip | **19.4** | **13.9** | **16.6** | **24.1** | **22.0** | **23.1** |

Table 4: Performance of the EPIC-KITCHENS-100 Multi-Instance Retrieval. Note that TBN † feature [14] is a combination of three modalities: RGB, Flow and Audio. Conversely, our approach only relies on RGB input. The `grey highlighted rows` correspond to **zero-shot evaluation**.

least one noun, (b) share at least one verb, and (c) share at least a verb-noun pair. Conversely, we define the following heuristics for negative sampling: (d) a random text-video pair from EgoClip, (e) a text-video pair from the same video, and (f) a text-video pair within 1 minute from the given video-text pair annotation timestamp. Tab. 3 shows that using solely verbs (a) or nouns (b) for positive selection degrades the accuracy performance with respect to naive InfoNCE. However, we successfully push the performance beyond the baseline results when considering both verbs and nouns jointly (c). Moreover, we notice that merely selecting negatives within the same video leads to better performance. In particular, we obtain the best performance for temporally "hard negatives" (f). Finally, we pick the optimal settings from positive and negative sides and combine them together for (g) EgoNCE and reach the best results.

## 6.3 Comparisons with State-of-the-arts

**Multi-Instance Retrieval.** In Tab. 4, we report both zero-shot and fine-tuning evaluation results. In the zero-shot setting, pretraining with EgoClip (3.8M), despite being smaller in scale, still outperforms CC3M+WebVid-2M (5.5M) and HowTo100M (136M), validating the unique benefit of pretraining on egocentric data. When fine-tuned with 4 frames (rows 5-9), EgoClip pretraining maintains a margin over the best baseline CC3M+WebVid-2M, further verifying the viewpoint domain gap within fine-tuning. Lastly, we increase the sample frames of our finalized model as well as the best competitor CC3M+WebVid-2M pretraining to 16 (rows 10-11). As expected, performance gains accompany the frame increase. We deem that notable benefits come from better temporal modeling for frequent action in the 1st-person view. Overall, our pretraining model outperforms the best baseline (JPoSE) by 1.0 mAP and 5.9% nDCG while requiring fewer frames and input modalities.

**Natural Language Query.** We report validation results on Tab. 5. We adopt the same baselines as introduced in [15], namely: 2DTAN [44] and VSLNet [45], and substitute the SlowFast-BERT features with our video and language representations. We observe a large boost in performance offered by our pretrained model on all metrics. Notably, we improve R@1 for IoU=0.3 from 5.45 to 10.84, despite our video branch not being pre-trained on Kinetics400. Besides, we significantly surpass VLP pretrained on CC3M+WebVid-2M and HowTo100M. We believe that this increase is due

| Methods | Video-text Pre-extrated Features | | IoU=0.3 | | IoU=0.5 | |
|---|---|---|---|---|---|---|
| | Vis-text Enc | Vis-text PT | R@1 | R@5 | R@1 | R@5 |
| 2D-TAN [44] | SlowFast+BERT | - | 5.04 | 12.89 | 2.02 | 5.88 |
| VSLNet [45] | SlowFast+BERT | - | 5.45 | 10.74 | 3.12 | 6.63 |
| VSLNet [45] | Frozen | HowTo100M | 3.95 | 8.72 | 2.01 | 4.62 |
| VSLNet [45] | Frozen | CC3M+WebVid-2M | 5.06 | 10.30 | 2.71 | 6.69 |
| VSLNet [45] | Frozen | EgoClip | 10.53 | 17.94 | 5.96 | 11.85 |
| VSLNet [45] | Frozen+EgoNCE | EgoClip | **10.84** | **18.84** | **6.81** | **13.45** |

Table 5: Recall for several IoUs on the NLQ task's val. set.

| Methods | Vis Enc | # Frames | Vis-Text PT | Train / FT Data | mAP (%) |
|---|---|---|---|---|---|
| Actor [46] | ResNet-152 | 25 | - | Charades-Ego (1st + 3rd) | 20.0 |
| SSDA [47] | I3D | 32 | - | Charades-Ego (1st + 3rd) | 23.1 |
| I3D [47] | I3D | 32 | - | Charades-Ego (1st). | 25.8 |
| Ego-Exo [48] | SlowFast (ResNet-101) | 32 | - | Charades-Ego (1st) | 30.1 |
| Frozen | TimeSformer | 16 | - | Charades-Ego (1st) | 28.8 |
| Frozen | TimeSformer | 16 | HowTo100M | Charades-Ego (1st) | 28.3 |
| Frozen | TimeSformer | 16 | CC3M+WebVid-2M | Charades-Ego (1st) | 30.9 |
| Frozen | TimeSformer | 16 | EgoClip | Charades-Ego (1st) | 31.2 |
| Frozen+EgoNCE | TimeSformer | 16 | EgoClip | Charades-Ego (1st) | **32.1** |
| Frozen | TimeSformer | 16 | HowTo100M | - | 9.2 |
| Frozen | TimeSformer | 16 | CC3M+WebVid-2M | - | 20.9 |
| Frozen | TimeSformer | 16 | EgoClip | - | 23.6 |
| Frozen+EgoNCE | TimeSformer | 16 | EgoClip | - | **25.0** |

Table 6: Performance of the action recognition on the Charades-Ego dataset (a first-person test set). The grey highlighted rows correspond to **zero-shot evaluation**.

to the egocentric data availability and the video-text interaction learned from large-scale pretraining. Please see Supplementary E.5 for the test set results.

**Action Recognition.** We conduct action recognition on Charades-Ego, where categories are short phrases like "Holding some clothes". Thus this task can be solved as a video-text retrieval by leveraging the text representation. We present the result in Tab. 6 under zero-shot and fine-tuning settings. In zero-shot settings, our model outperforms two supervised baselines, which validates the stronger generalization of jointly learning video-text features. After fine-tuning (rows 5-9), our model surpasses all VLP counterparts and improves over the state-of-the-art classifier Ego-Exo by 2.0% with fewer sampled frames, which shows the superior advantage of joint video-text representations.

| Methods | Video Pre-extracted Features | | IoU=0.3 | | IoU=0.5 | | IoU=0.7 | | mAP (%) @ IoU | | | |
|---|---|---|---|---|---|---|---|---|---|---|---|---|
| | Vis Enc | Vis-text PT | R@1 | R@5 | R@1 | R@5 | R@1 | R@5 | 0.1 | 0.3 | 0.5 | Avg |
| VSGN [49] | SlowFast | - | 33.45 | 58.43 | 25.16 | 46.18 | 15.36 | 25.81 | 9.10 | 5.76 | 3.41 | 6.03 |
| VSGN [49] | Frozen | HowTo100M | 31.40 | 52.61 | 22.28 | 41.29 | 13.41 | 23.21 | 9.83 | 6.72 | 3.84 | 6.72 |
| VSGN [49] | Frozen | CC3M+WebVid-2M | 32.08 | 56.40 | 23.46 | 43.81 | 13.73 | 23.77 | 9.83 | 6.40 | 3.86 | 6.58 |
| VSGN [49] | Frozen | EgoClip | 40.06 | 63.71 | 29.59 | 48.32 | 17.41 | 26.33 | 15.90 | 10.54 | 6.19 | 10.69 |
| VSGN [49] | Frozen+EgoNCE | EgoClip | **40.43** | **65.67** | **30.14** | **51.98** | **19.06** | **29.77** | **16.63** | **11.45** | **6.57** | **11.39** |

Table 7: Recall and mAP metrics for several IoUs on the Moment Query task's val. set.

**Moment Query.** This task investigates the quality of video-only features. We extract video features and provide them as input to the VSGN model [49]. We report the validation results in Tab. 7, We find that our features achieves the best performance over SlowFast features with an increase of 4.66% in Avg mAP. Moreover, we maintain better performance with respect to 3rd-person large-scale pretraining datasets. This demonstrates that the 1st-person VLP model also learns competitive video representations. Please see the Supplementary E.6 for the test set results.

**Object State Change Classification.** We report the validation results on Tab. 8. Once again, our model achieves the best performance of all baselines, 2.4% than CC3M+WebVid-2M counterparts, which indicates our visual representations are able to focus on the fine-grained clues related to state changes.

**Summary of EgoNCE.** From the above experimental results, Frozen pretrained on EgoClip with the EgoNCE objective brings a consistent improvement over the InfoNCE on all downstream tasks, which comprehensively demonstrates the effect of EgoNCE, as well as the decision from EgoMCQ.

| Methods | Vis-Text PT | Acc. (%) |
|---|---|---|
| Always Positive | - | 48.1 |
| Bi-d LSTM [50] | ImageNet | 65.3 |
| I3D (ResNet-50) [51] | - | 68.7 |
| Frozen | - | 70.3 |
| Frozen | HowTo100M | 71.7 |
| Frozen | CC3M+WebVid-2M | 71.5 |
| Frozen | EgoClip | 73.4 |
| Frozen+EgoNCE | EgoClip | **73.9** |

Table 8: Accuracy metric on the Object State Change Classification task's val. set.

# 7 Conclusion, Limitations, and Societal Impacts.

To the best of our knowledge, this work is the pioneering work to unlock Egocentric VLP. **(i)** We devise a principled data curation and create EgoClip, an egocentric large-scale text-video pretraining dataset with 3.8M clip-text pairs well-chosen from Ego4D. **(ii)** We exploit the particular characteristics of egocentric videos and devise EgoNCE with meaningful sampling strategies for effective egocentric pretraining. **(iii)** We create EgoMCQ, an egocentric video-language benchmark close to the pretraining set to support efficient exploration and development of EgoClip and EgoNCE. Finally, we further demonstrate the strong representation of our egocentric pretraining on five tasks across three datasets. We believe that our EgoClip, EgoMCQ and EgoNCE would greatly benefit the egocentric video community, laying a good foundation for the new research trend of egocentric VLP. **Limitations.** Our pretraining approach does not take into account the long-term temporal dependencies in long Ego4D videos. We leave this for future work.
**Societal impact.** Egocentric VLP learns real-world perception knowledge that may contribute to practical applications such as augmented reality and robotics. However, Ego4D videos collected by participants may contain users' privacy and unintended biases, so should be used cautiously. We refer the readers to the Ego4D paper about further privacy and societal impacts.

# 8 Acknowledgements

This project is supported by the National Research Foundation, Singapore under its NRFF Award NRF-NRFF13-2021-0008, and Mike Zheng Shou's Start-Up Grant from NUS. The computational work for this article was partially performed on resources of the National Supercomputing Centre, Singapore. Michael Wray and Dima Damen are supported by EPSRC UMPIRE (EP/T004991/1). Mattia Soldan and Bernard Ghanem are supported by the King Abdullah University of Science and Technology (KAUST) Office of Sponsored Research through the Visual Computing Center (VCC) funding, as well as, the SDAIA-KAUST Center of Excellence in Data Science and Artificial Intelligence (SDAIA-KAUST AI). Thanks to Tencent Data Platform for the support of computing resources. Our work is built upon the Ego4D dataset, and we greatly appreciate the contributions and efforts of the Ego4D community.

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
