## Appendix of EgoVLP

We present the following items in the supplemental material:

- **A**. Differentiating Egocentric VLP and Ego4D in Sec. A.
- **B**. Construction details and statistics of EgoClip pretraining dataset in Sec. B.
- **C**. Construction details and statistics of EgoMCQ benchmark in Sec. C.
- **D**. Technical details of our VLP model in Sec. D.
- **E**. Additional experimental details and results in Sec. E.

## A  Differentiating Egocentric VLP and Ego4D

In our work, we study the video-language pretraining in a specific yet significant domain - the 1st-person view, which is motivated by the release of the Ego4D dataset. However, there is a long way to pave from the Ego4D dataset to Egocentric VLP, which consists of the pretraining dataset, development set, model designs, and transferability evaluation. Since they are not as fully explored as their third-person counterparts, thus we pioneer them by ourselves and conduct *a systematic study toward the egocentric video-language pretraining* - the contribution of our work.

### A.1  Pretraining dataset

Despite the merits of Ego4D, it has not been proposed for video-language pretraining, and cannot be directly used as its untrimmed videos, no direct video-text pairs, and noisy data. We thus see our clear distinction and contribution in proposing a successful approach to curate a pretraining dataset, our proposed EgoClip. Notably, It is also non-trivial to figure out what is the best way of curating Ego4D to create a pretraining dataset EgoClip, e.g., our pairing approach outperforms the naive strategy with a large margin in the development set, which requires substantial design and experimental validations. We add a Tab. 9, as an extension of Tab. 1, to clearly show their difference.

| Dataset | Ego? | Domain | Dur (hrs) | # Clips | # Texts |
|---|:---:|:---:|:---:|:---:|:---:|
| Ego4D [15] (untrimmed) | ✓ | **diverse** | **3.6K** | - | **5.0M** |
| **EgoClip** (well-curated from Ego4D) | ✓ | **diverse** | $2.9K$ | **3.8M** | $3.8M$ |

Table 9: Comparison of EgoClip and Ego4D dataset.

### A.2  Development set

In the 1st-person domain, there is lacking a satisfactory benchmark that good aligns with pretraining data diversity and focuses on video-text alignment. Therefore, we propose a new development set i.e. EgoMCQ to power rapid design of video-text pretraining i.e. its pretraining dataset and model pretraining objective.

### A.3  Model designs

We select Frozen as the baseline because its elegant and scalable dual-encoder architecture is representative in state-of-the-art VLP methods. Besides, corresponding to MIL-NCE [23] built on top of the 3rd-person domain's HowTo100M [10], we aim to explore a general pretraining objective i.e., EgoNCE to learn rich video-text representations in 1st-person domains.

### A.4  Transferability evaluation

Extensive experiments and promising results demonstrate the effectiveness and necessity of Egocentric VLP, which will greatly benefit the egocentric community. Note that Ego4D has not been used previously for any downstream tasks on other datasets. This is also where our work makes significant value.

# B  Construction details and statistics of EgoClip pretraining dataset

## B.1  Data filtering

After we source video-text data for EgoClip, we adopt the following criteria to further reduce noise:

**(i)** We select double-sized stereo videos (1.3% videos dur) and keep half per video for a normal size.

**(ii)** We discard videos with an aspect ratio greater than 2 (0.4% videos dur).

**(iii)** We filter narrations with unsure tags (4.0% texts) e.g. "`#C C washes #unsure in sink`".

**(iv)** We remove narrations less than 3 words (0.9% texts), since such narrations generally cannot be deduced from the video, e.g., "`#C C speaks`", "`#C C looks`".

## B.2  Data compression

The Ego4D videos are untrimmed, which tend to be very long (average 24 mins and max to 7 hrs) and have large resolution (e.g., $1920 \times 1080$, $1440 \times 1080$), so it is impossible to adopt untrimmed videos as model input due to heavy data loading. Therefore we propose to compress them:

**(i)** We first resize all videos with short size 256.

**(ii)** Chunk each resized video into several segments, which are up to 10 min in length.

During pretraining, given the start and end time points of a clip, we only load the segment that this clip belongs to, rather than the whole video. To this end, we are able to perform efficient end-to-end pretraining with raw RGB videos as model input. One epoch of pretraining 3.8M video-text pairs costs 6 hrs on 32 V100 GPUs (192 GPU hrs).

## B.3  Data analysis

**Geographic diversity.** We present the distribution of EgoClip clips source in Fig. 3, which covers worldwide 13 institutions from 9 different countries [15], including: Europe (UK, Italy); Asia (India, Japan, Singapore, Kingdom of Saudi Arabia); America (USA, Colombia); Africa (Rwanda). Therefore, our created pretraining dataset inherited the good geographic as well as participants diversities of Ego4D (More details can be found in "Supp. C. Demographics" in Ego4D paper [15]).

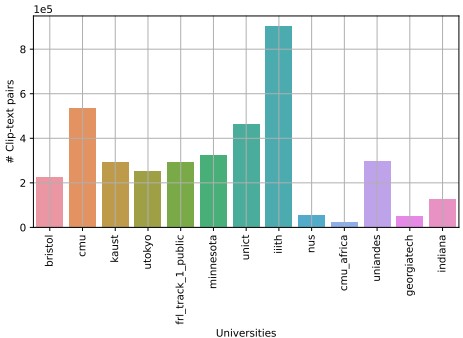

Figure 3: Institution distribution of EgoClip

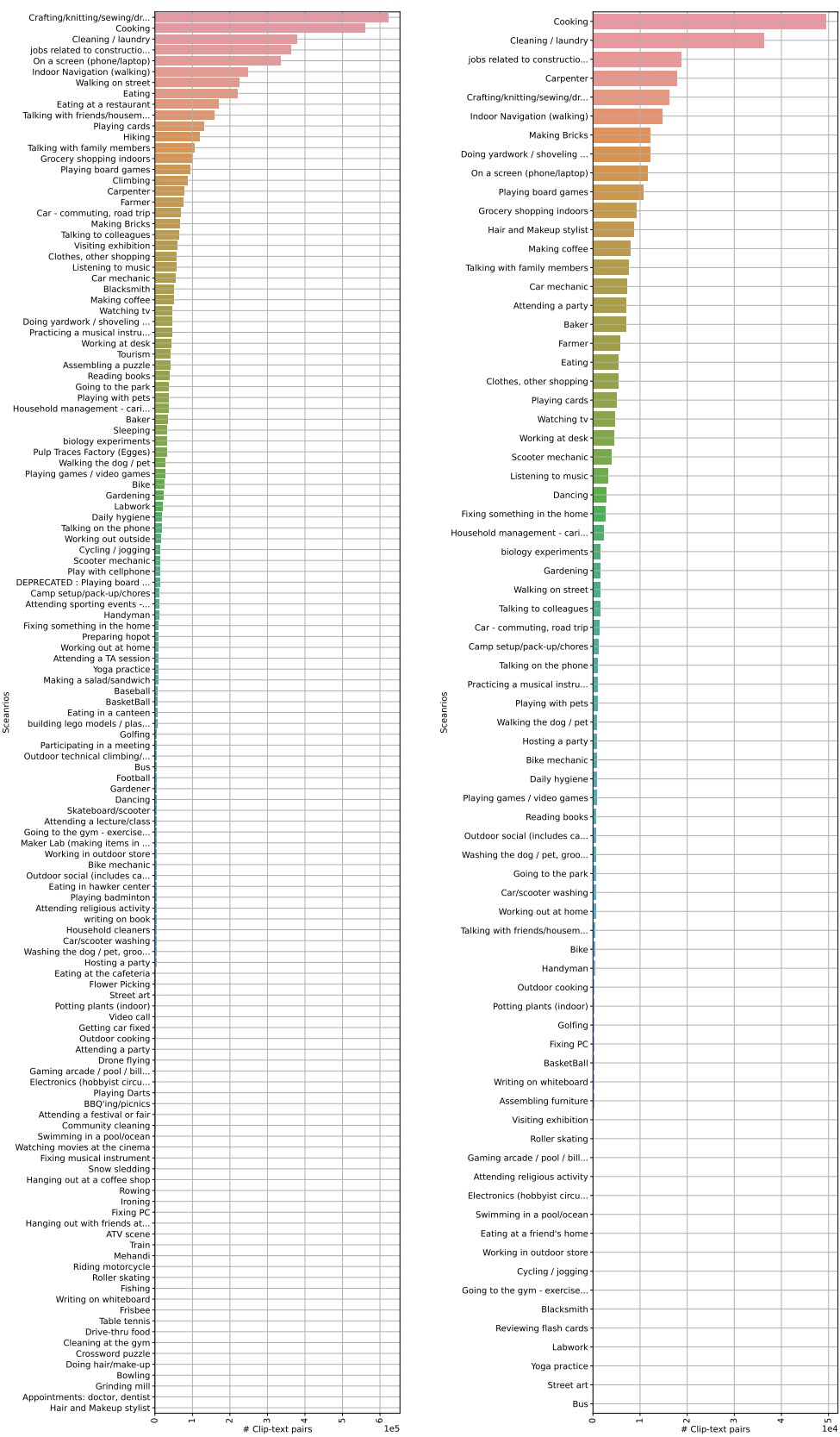

Figure 4: Scenario distribution of EgoClip        Figure 5: Scenario distribution of EgoMCQ

**Scenario diversity.** We have statistics the scenario distribution of EgoClip in Fig. 4, which covers 129 human daily scenarios e.g., household (cooking, cleaning), outdoor (shopping, hiking), workplace (at desk, on a laptop), leisure (playing board games), etc. Notably, this distribution is long tailed, where the largest scenario "Crafting/knitting/sewing/drawing/painting" includes 622K (11.1%) and the smallest scenario "Hair and Makeup stylist" contains 35 instances.

**Clip analysis.** We present the statistics on the created clips in EgoClip. Fig. 6 (a) shows the distribution of clip frequency over the 2.9K pretraining set videos (For each video, we calculate two frequencies from two annotators respectively). The varying clip frequencies are mainly dependent on manual narrations that are annotated based on the video scenarios and activities. There have average 13.4 clips per minute of video, maximize to 175.8 narrations / minute and minimize to 0.06 narrations / minute. Our clip creation strategy Eq. (1) takes this characteristic into account by estimating clip length based on the frequency of the video that the clip belongs. Fig. 6 (b) displays the distribution of clip duration. The average duration is 0.98 seconds with a standard deviation of 0.95 seconds, and 69.5% of clips are less than 1.0 seconds in length, due to the massive atomic instantaneous actions densely labeled by Ego4D. Besides, the clip might be max to 65.36 seconds, which corresponding to the scenario that "a people walking in a forest".

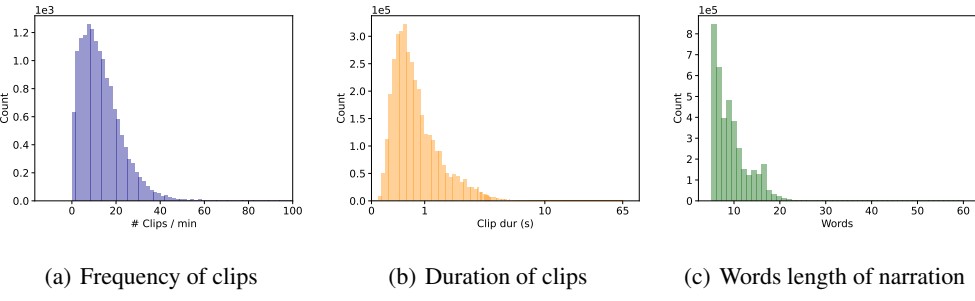

(a) Frequency of clips      (b) Duration of clips      (c) Words length of narration

Figure 6: Clip and narration distribution of EgoClip

**Narration analysis.** In Fig. 6 (c), we present the distribution of narration words length. The average words length of EgoClip narration is 9.39. Notably, the EgoClip narrations cover 116 verbs and 555 nouns, where we merge the semantically synonyms words, e.g., the nouns of "handkerchief","napkin","serviette","tissue","wipe" both belong to "napkin". Each narration of EgoClip have 1.84 nouns and 0.87 verbs on average.

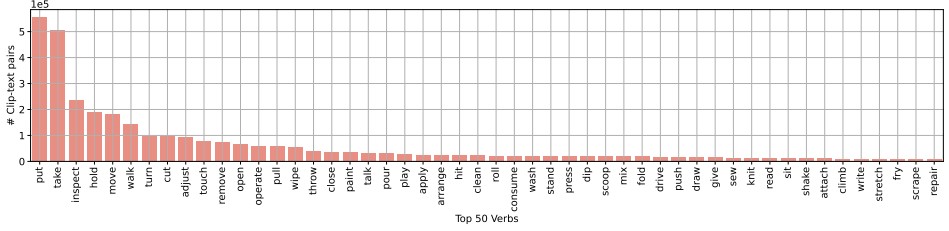

(a) Top 50 most frequently verbs distribution

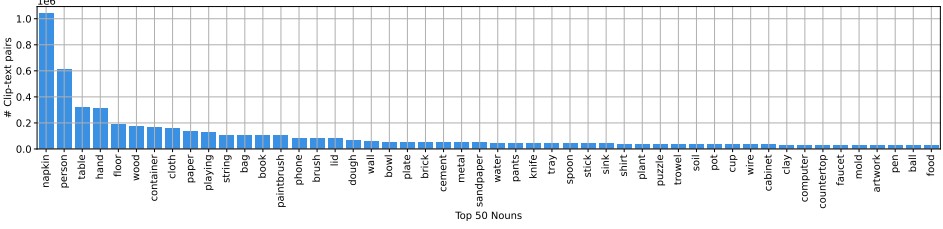

(b) Top 50 most frequently nouns distribution

Figure 7: Verbs and nouns distributions of EgoClip's narrations

We further display the distribution of the top 50 most frequently verbs and nouns of EgoClip in Fig. 7. The most common nouns is "napkin", which appeared in 1.0M (27.06%) clips.

**Visualizations.** In Fig. 8, we visualize some clip-text pairs created by our strategy.

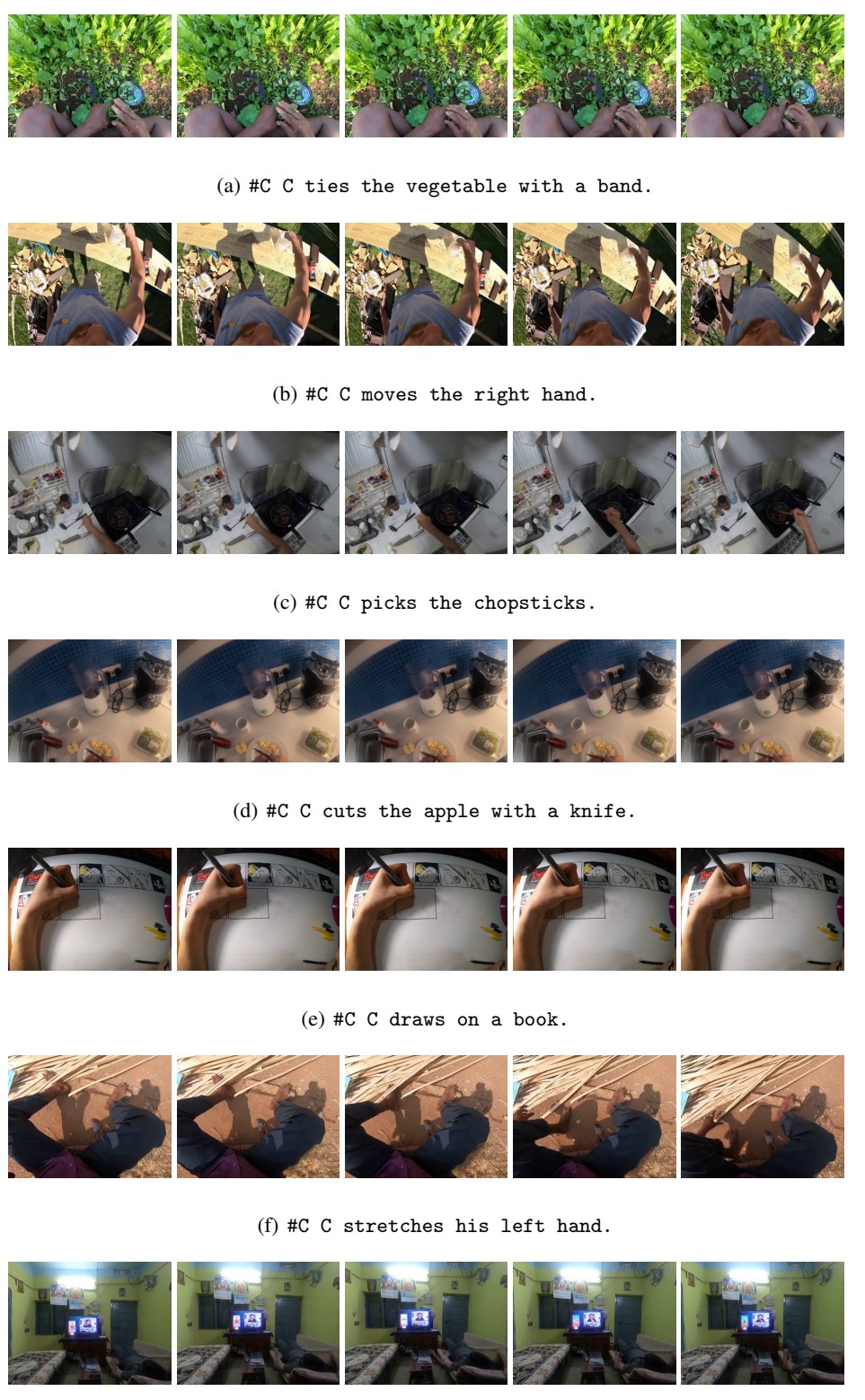

(a) #C C ties the vegetable with a band.

(b) #C C moves the right hand.

(c) #C C picks the chopsticks.

(d) #C C cuts the apple with a knife.

(e) #C C draws on a book.

(f) #C C stretches his left hand.

(g) #O A man X moves hand from the table.

Figure 8: Visualization of EgoClip clip-text pairs. We sample five frames uniformly for each clip and take its narration as its caption.

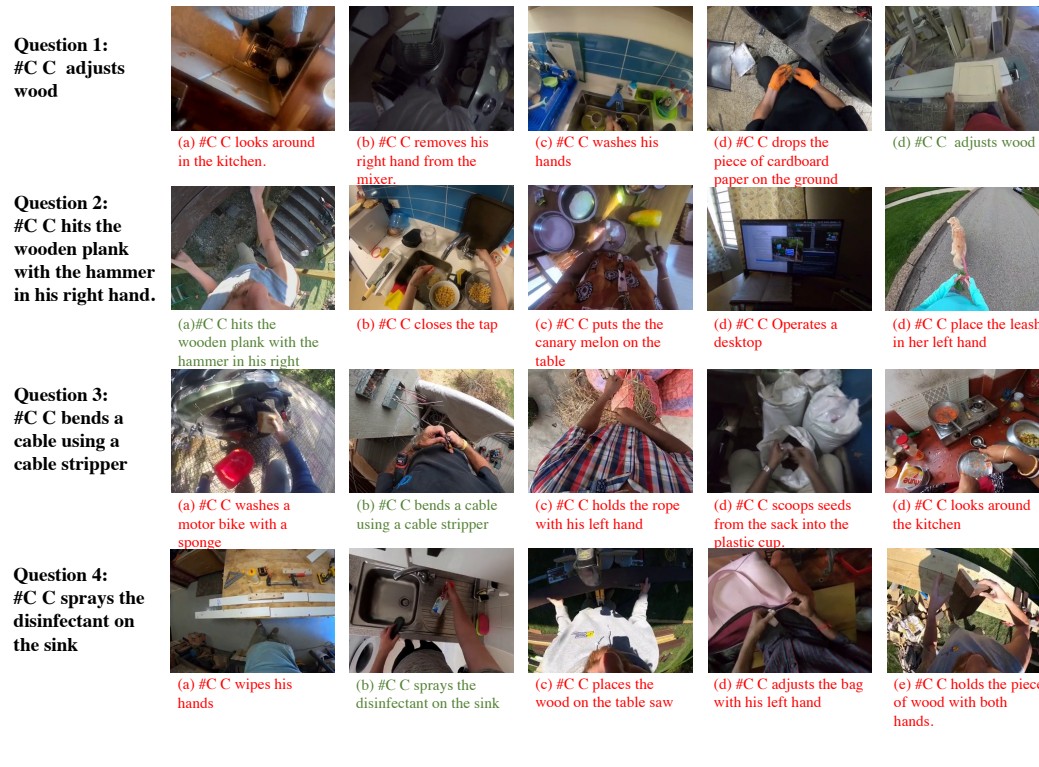

(a) Inter-video setting

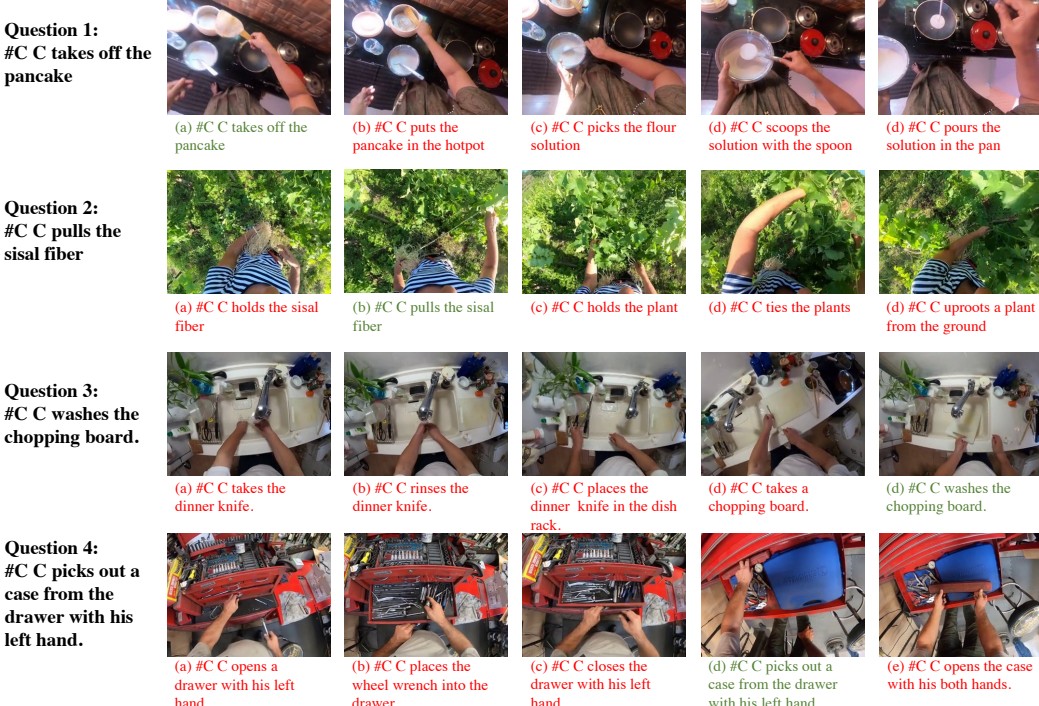

(b) Intra-video setting

Figure 9: Visualization of EgoMCQ under two settings. Left are the text questions; Right are the five candidate clips for each question and the text below as clip's narrations. The correct clip's narrations is highlighted in green and the wrong in red.

# C Construction details and statistics of EgoMCQ benchmark

## C.1 Data deduplication

To ensure repetitions do not appear in five options, we devise a deduplication strategy. Initially, we use Sentence-BERT to extract sentence-level embeddings of narrations and set a manual threshold to remove repetitions. But in this way, it is hard to control the fine-grained diversity between narrations, e.g., two narrations "`#C C closes the refrigerator with his left hand.`" and "`#C C opens the refrigerator with his left hand.`" only differ in one word. These two sentences have a high score in sentence-level similarity, but are entirely different in semantic meanings. We hope to keep them and let the model distinguish them, especially in our intra-video setting.

Therefore, we propose to extract the first verb and the first noun of each narration and use them to define a tag for each narration. The narrations shared with the same verb and the noun will be assigned the same tag. We also consider the words synonyms (based on Ego4D taxonomy dictionary [15]). For instance, "`#C C take the phone`" and "`#C C pick the cellphone`" are semantically same in verb and noun thus will be assigned the same tag. Then the narrations shared with the same tag are treated as repetitions, we only keep one of them and sample a new one until the tags of the five options are different.

## C.2 Multiple-views removing

We first select videos from NUS/Minnesota/Georgia Tech/Indiana sources, which contribute to the multi-camera video data. Then, based on the metadata of the video (i.e. times when videos were collected), we observed that videos collected in the same timeframe tend to be multi-views of the same recording, so we manually group these videos into the same split to ensure the same scene does not appear in another split.

## C.3 Data analysis

We finalize 39K questions covering 198K narrations with 468 hours of video, where the "inter-video" has 24K questions covering 290.3 hours of videos. And the "intra-video" has 15K questions and covers 178.3 hours of videos. The average duration among the five options is 34.2 seconds.

**Geographic diversity.** We present the geographic diversity of EgoMCQ in Fig. 10, which covers 13 institutions and is align with the geographic diversity of EgoClip.

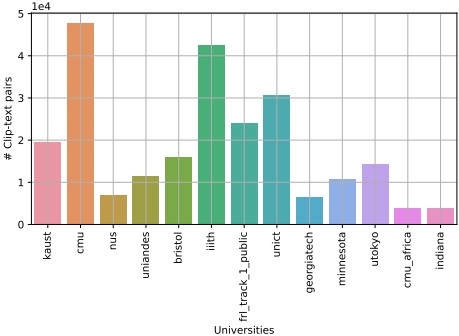

Figure 10: Institution distributions of EgoMCQ

**Scenario diversity.** In Fig. 5, we present the scenario distribution of EgoMCQ, which covers 74 scenario. The largest scenario "Cooking" includes 49K (15.3%) clips and the smallest scenario "Bus" contains 6 instances. EgoMCQ covers 71% of scenarios in EgoClipand has other 3 scenarios not appear in EgoClip. EgoMCQ is close to EgoClip both in terms of geography and scene diversity, making it a good development set for EgoClip pretraining.

**Verbs and Nouns.** EgoMCQ covers 198K narrations and each narration contains 3.15 nouns and 0.97 verbs in average. In Fig. 11, we display the top 50 most frequently verb and nouns of EgoMCQ. The mostly common noun is "hand", covering 86K (36.2%) instances and the mostly frequently verb is "pick", which covers 28K (12.0%) instances.

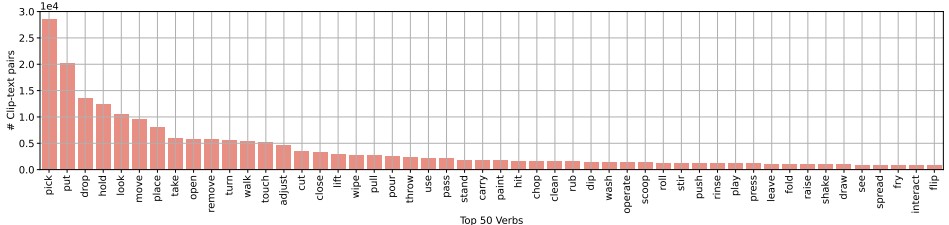

(a) Top 50 most frequently verbs distribution

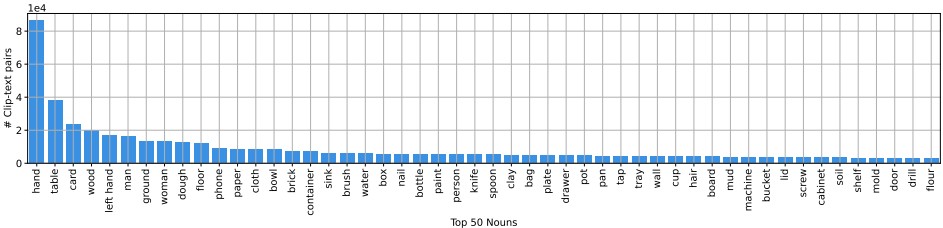

(b) Top 50 most frequently nouns distribution

Figure 11: Verbs and nouns distributions of EgoMCQ's narration.

**Visualization.** In Fig. 9, we display examples of both the intra and inter settings of EgoMCQ.

## D  Technical details of our VLP model

In this section, we present more technical details of our VLP model, mainly architecture and pretraining objective.

### D.1  Architecture: Frozen-in-time [3]

**Video Encoder.** The video encoder is built upon Timesformer [32], a convolution-free video backbone that divides space-time attention in an efficient manner. Take a RGB video clip $\mathcal{V}_i \in \mathbb{R}^{T \times 3 \times H \times W}$ with $T$ frames and resolution $H \times W$ as input, the clip is first divided into $M \times N$ patches $\mathbf{p} \in \mathbb{R}^{M \times N \times 3 \times P \times P}$ with size of $P \times P$, where $N = HW/P^2$. Next, patches $\mathbf{p}$ are linearly embed as a token sequence $\mathbf{z} \in \mathbb{R}^{MN \times D}$ with $D$ dimension. Then, the learned temporal embeddings $E^s \in \mathbb{R}^{N \times D}$ and spatial positional embeddings $E^t \in \mathbb{R}^{N \times D}$ are added to each input token. Besides, a learnable CLS token is concatenated at the beginning of the token sequence. Finally, these token sequences are fed into Timesformer and output the CLS token of the last block, which is further projected to a $d$ dimension embedding by a linear layer to form the final clip representation $\mathbf{v}_i \in \mathbb{R}^d$.

**Text Encoder.** The text encoder is built upon DistillBERT [33], which has $40\%$ less parameters than BERT while also preserves over $95\%$ performance, thus is efficient. Taking a sentence $\mathcal{T}_i$ as input, first tokenize it as a sequence of tokens and feed it into DistillBERT. Similar to the video encoder, the CLS token of DistillBERT's output is projected as $\mathbf{t}_i \in \mathbb{R}^d$ for the final text representation.

### D.2  Pretraining objective: EgoNCE

To supplement the Eq. 2 and Eq. 3, we first formulate the complete form of InfoNCE:

$$
\begin{aligned}
\mathcal{L} &= \mathcal{L}_{\text{v2t}} + \mathcal{L}_{\text{t2v}} \\
&= -\frac{1}{|\mathcal{B}|} \sum_{i \in \mathcal{B}} \log \frac{\exp(\mathbf{v}_i^T \mathbf{t}_i / \tau)}{\sum_{j \in \mathcal{B}} \exp(\mathbf{v}_i^T \mathbf{t}_j / \tau)} - \frac{1}{|\mathcal{B}|} \sum_{i \in \mathcal{B}} \log \frac{\exp(\mathbf{t}_i^T \mathbf{v}_i / \tau)}{\sum_{j \in \mathcal{B}} \exp(\mathbf{t}_i^T \mathbf{v}_j / \tau)}
\end{aligned}
\tag{4}
$$

and our EgoNCE extends the above as Eq. 5 via two sampling strategies:

$$\mathcal{L}^{\text{ego}} = \mathcal{L}^{\text{ego}}_{\text{v2t}} + \mathcal{L}^{\text{ego}}_{\text{t2v}}$$
$$= -\frac{1}{|\widetilde{\mathcal{B}}|} \sum_{i \in \widetilde{\mathcal{B}}} \log \frac{\sum_{k \in \mathcal{P}_i} \exp(\mathbf{v}_i^T \mathbf{t}_k / \tau)}{\sum_{j \in \mathcal{B}} \left( \exp(\mathbf{v}_i^T \mathbf{t}_j / \tau) + \exp(\mathbf{v}_i^T \mathbf{t}_{j'} / \tau) \right)} \tag{5}$$
$$- \frac{1}{|\widetilde{\mathcal{B}}|} \sum_{i \in \widetilde{\mathcal{B}}} \log \frac{\sum_{k \in \mathcal{P}_i} \exp(\mathbf{t}_i^T \mathbf{v}_k / \tau)}{\sum_{j \in \mathcal{B}} \left( \exp(\mathbf{t}_i^T \mathbf{v}_j / \tau) + \exp(\mathbf{t}_i^T \mathbf{v}_{j'} / \tau) \right)}.$$

For positive sampling (the numerator term), we pre-extract the nouns and verbs for each narration $\mathcal{T}_i$ before pretraining and define two word vectors $\mathbf{w}_i^n \in \{0,1\}^{K_1}$ and $\mathbf{w}_i^v \in \{0,1\}^{K_2}$ to encode the appearing nouns and verbs in sentence, where $K_1$ and $K_2$ denote the number of nouns and verbs in EgoClip (Refer to Sec B narration analysis). During pretraining, for another instance $j$ within batch, we calculate the $s_{ij} = (\mathbf{w}_i^n)^T \mathbf{w}_j^n \cdot (\mathbf{w}_i^v)^T \mathbf{w}_j^v$, if $s_{ij} > 0$, we regard instance $j$ is one of the positive sample $j \in \mathcal{P}_i$ of instance $i$. Notably, the positive sampling space $\mathcal{P}$ would cover $\widetilde{\mathcal{B}}$ when working with the negative sampling strategy.

For negative sampling (the denominator term), each time we sample an instance $i$, we sample an instance $i' \in \mathcal{V}_i$ in the same video and close in time (less than 1 min) to generate the negative sample $i' \in \mathcal{N}(i)$ of instance $i$. Notably, in this way, the actual instance within the batch $|\widetilde{\mathcal{B}}| = 2N$ will be double the batch size $|\mathcal{B}| = N$. In practice, we have to halve the batch size due to GPU memory limitations. Under halving the batch size, random sampling doesn't help in our method, which can be concluded by comparing baseline InfoNCE and variants (d) in Tab. 3 of the main body, where the batch size of the latter is half of the former. Despite this, our proposed sampling strategy (f) can successfully improve the pretraining effect beyond baseline.

In contrast to the conventional negative sampling from the same video [37, 52], we specifically design our temporally adjacent negative sampling strategy to focus on the frequent appearance changes in egocentric videos, which has not been explored in previous approaches.

# E  Additional experimental details and results

## E.1  Implementation details

Following the settings of official Frozen [3], the video encoder is initialized with ViT [53] weights trained on ImageNet-21K with sequence dimension $D = 768$. The text encoder is based on huggingface's `distilbert-base-uncased`. The dimension of common feature space is set as 256, and the temperature parameter is set to 0.05. During pretraining, each video is resized to $224 \times 224$ as input with sample frames number 4 and batch size 512. We use the Adam optimizer with a learning rate of $3 \times 10^{-5}$ with a total epoch of 10. When transferring to downstream tasks, we select the checkpoints with the best score on EgoMCQ benchmark i.e. average accuracy of inter-video and intra-video settings by default.

## E.2  Downstream settings

We present the setting details of the downstream tasks we evaluated. For a fair comparison, for VLPs variants pretrained on different datasets, we use the same settings on downstream tasks, such as the fine-tuning objective.

**EPIC-KITCHENS-100 Multi-Instance Retrieval.** In this task, after we finalize video-text pretraining, we continue to fine-tune the VLP model and keep most settings of pretraining (e.g., input resolution, learning rate). Notably, we set the training epoch as 100 and replace the training objective as Multi-instance Maxmargin loss in Eq. 6, which is same as the baseline method JPoSE [43]. The reason for this is that in this task a narration may be jointly associated with multiple clips, so multi-instance learning mechanism can better handle such a situation. And this dataset also provides the action label to calculate the correlation $c_{ij}$ between two clip-text pairs $(i,j)$, which supports the implementation of Multi-instance Maxmargin loss.

$$\mathcal{L} = \sum_{(i,j,k) \in \Omega} \max \left( \gamma + \mathbf{v}_i^T \mathbf{t}_j - \mathbf{v}_i^T \mathbf{t}_k \right) + \left( \gamma + \mathbf{t}_i^T \mathbf{v}_j - \mathbf{t}_i^T \mathbf{v}_k \right), \tag{6}$$

where $\Omega = \{(i, j, k) \,|\, j \in i^+, k \in i^-\}$ is a triple, which indicates a positive instance $j$ and a negative instance $k$ for $i$. In our setting, we define the positive set as $i^+ = \{j|c_{ij} > 0.1\}$ and the negative as the remains sample within batch. The $\gamma$ is a margin factor and we set it as $0.2$.

**Charades-Ego Action Recognition.** In this task, the textual categories are short phrases like "Holding some clothes". Thus, we regard this task as a kind of video-text retrieval by leveraging the text representation and using the InfoNCE as fine-tuning objective. We set the epoch number as 10 and keep other parameters unchanged.

**Ego4D Natural Language Query** This task is a kind of video-text localization and is hard to perform end-to-end training (since a clip might long to 1200 seconds). The baseline method [45] takes 2304 dim SlowFast features (1.87 fps, with Kinetics 400 pretrained) and 768 dim BERT features as input. Therefore, we propose to replace the baseline input features as features of pretrained VLP video and text encoders to evaluate the pretraining effectiveness. We extract the features with the same fps 1.87 and sampling frame number 4. In fine-tuning stage, we keep the default setting of [45].

**Ego4D Moment Query** This task is a video-only task: temporal action localization. Similar to Natural Language Query task, we replace the input Slowfast features of baseline VSGN [49] with VLP video features for evaluation. The extraction details are the same as Natural Language Query.

**Ego4D Object State Change Classification** This is an action classification task, we sample each clip with 16 frames as input and use the cross-entropy as fine-tuning objective. The epoch is set as 10.

### E.3  VLP Evaluation on EgoMCQ

In Tab. 10, we display EgoMCQ evaluation result of Frozen pretrained on different video-text datasets.

| VL Pretraining | Accuracy (%) | |
|---|---|---|
| | Intra-video | Inter-video |
| Random | 20.0 | 20.0 |
| EPIC-KITCHENS-100 | 28.1 | 22.7 |
| HowTo100M | 31.5 | 21.6 |
| CC3M+WebVid-2M | 62.5 | 27.4 |
| EgoClip | 89.4 | 51.5 |
| EgoClip w/ EgoNCE | 90.6 | 57.2 |

Table 10: Results of VLPs pretrained on different datasets in EgoMCQ

As shown, pretraining with EPIC-KITCHENS-100 dataset (1st-person view, 67.2K pairs) reach comparable performance with HowTo100M pretraining (3rd-person view, 136M noisy pairs), which demonstrates the major domain gaps. Besides, Frozen with CC3M+WebVid-2M pretraining reach significant improvement on the intra-video setting, but minor in inter-video. We speculate this due to CC3M+WebVid-2M dataset covering a wide range of appearance information but still less exploration in the fine-grained action e.g. human-object interaction.

### E.4  Training Curves of EPIC-KITCHENS-100 video-text retrieval

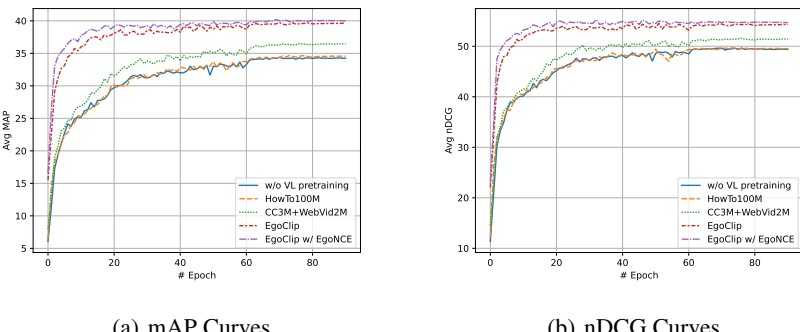

(a) mAP Curves          (b) nDCG Curves

Figure 12: Training Curves of EPIC-KITCHENS-100 video-text retrieval

In Fig. 12, we display training curves of EPIC-KITCHENS-100 video-text retrieval under different video-text pretraining, which also includes a baseline without video-text pre-training. We can found that: Variants with video-text pretraining have a faster rise in performance. Except for HowTo100M, which is similar to variant without video-text pretraining. Especially with EgoClip for egocentric pretraining, the VLP model achieves nearly convergent performance with only a small number of epochs (less than 20). With EgoNCE as pretraining objective, this positive effect is further enhanced.

## E.5 Results on test set of Natural Language Query

In Tab. 11, we found the similar conclusions in test set of Natural Language Query, pretraining with EgoClip and EgoNCE reach the optimum performance.

| Methods | Video-text Pre-extrated Features | | IoU=0.3 | | IoU=0.5 | |
| | Vis-text Enc | Vis-text PT | R@1 | R@5 | R@1 | R@5 |
|---|---|---|---|---|---|---|
| VSLNet | SlowFast+BERT | - | 5.47 | 11.21 | 2.80 | 6.57 |
| VSLNet | Frozen | HowTo100M | 3.77 | 6.87 | 1.62 | 3.45 |
| VSLNet | Frozen | CC3M+WebVid-2M | 4.87 | 8.67 | 2.50 | 4.97 |
| VSLNet | Frozen | EgoClip | 10.34 | 15.81 | 6.24 | 10.39 |
| VSLNet | Frozen+EgoNCE | EgoClip | **10.46** | **16.76** | **6.24** | **11.29** |

Table 11: Recall for several IoU on the NLQ task's test set.

## E.6 Results on test set of Moment Query

We further display the test set results of Moment Query in Tab. 12, pretraining with EgoClip and EgoNCE reach the best performance, $3.78\%$ on R@1 and $4.65\%$ on Avg mAP over the baseline.

| Methods | Video-text Pre-extrated Features | | IoU=0.5 | mAP(%)IoU |
| | Vis-text Enc | Vis-text PT | R@1 | Avg |
|---|---|---|---|---|
| VSGN | SlowFast | - | 24.25 | 5.68 |
| VSGN | Frozen | HowTo100M | 18.06 | 5.28 |
| VSGN | Frozen | CC3M+WebVid-2M | 19.74 | 5.95 |
| VSGN | Frozen | EgoClip | 27.98 | 9.78 |
| VSGN | Frozen+EgoNCE | EgoClip | **28.03** | **10.33** |

Table 12: Recall and mAP metrics on the MQ task's test set.

## E.7 Visualization

To intuitively understand the effect of egocentric pre-training, in Fig. 13, we compare the EPIC-KITCHENS-100 video-text retrieval results between our pre-training (EgoClip w/ EgoNCE) and CC3M+WebVid-2M pre-training, both fine-tuning with 16 frames. The numbers after each narration represent the correlation scores between the query and the retrieval result, with 1 being the best.

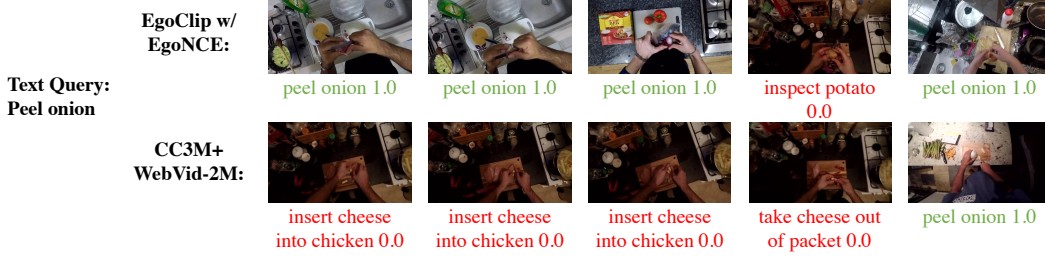

Figure 13: Visualization of EPIC-KITCHENS-100 video-text retrieval. Given the same text query, we compare the **Top-5 results** of 1st-person pretraining and 3rd-person pretraining.