# OpenReview forum: "Egocentric Video-Language Pretraining"
_NeurIPS.cc/2022/Conference — NeurIPS 2022 Accept_

### Official Review · Reviewer_pWxJ · 2022-07-08

**Rating:** 5
**Confidence:** 4
**Soundness:** 3 good
**Presentation:** 3 good
**Contribution:** 3 good

**Summary:**

This paper proposes a new benchmark for egocentric video understanding based on Ego4D, a pre-training set, and a model for the pre-training and downstream fine-tuning. Given the untrimmed Ego4D videos, the authors do the curations and build the EgpClip. Based on the recent work Frozen, a contrastive video learning model with a new loss RgoNCE is proposed. To test the above design, the authors also propose a valid set named EgoMCQ based on the val data of Ego4D. On several downstream tasks, the pre-training set and model show impressive performance compared to existing first-view pre-training works given the large-scale video-text data from Ego4D.

**Questions:**

L184-185: to assure that the scene is not visible during pretraining, we manually remove videos that share multiple views with the videos in EgoClip.
Any details about the manually removing?

**Limitations:**

N/A.

**Strengths And Weaknesses:**

Pros:
+ The curation and cleaning for Ego4D are important and beneficial for the community.

+ The illustrations are detailed and easy to follow. The suppl gives much useful information.

+ The experiments indeed show good signs when we use this new curated dataset for ego-centric video learning.

+ The main setting and curation are sound.

Cons:
- The largest concern: this work seems not to differentiate the contribution of Ego4D and this submission.

1. The tab1 is misleading, as the data scale and contribution belong to Ego4D instead of this work. This work conducts mainly the curation and cleaning. So I suggest comparing both Ego4D and EgoClip here to avoid this misleading.

2. The results look good and impressive, but the readers may wonder how much of them come from the curation? Though without curation, the videos in Ego4D may be not easy to directly use, the performance improvements can be definitely thanks to the data contribution of Ego4D when compared with the first-view data pre-training. Only an "EgoClip" is weird and misleading. Thus, I suggest the authors add the tests to use the original videos of Ego4D in soma ways, or if not at least discuss this clearly. If we think about this point, the data contribution of this work and the performance claim may be misaligned.

3. Considering the method contribution is relatively smaller compared to the data,  in my opinion, this work may be more suitable for the benchmark and dataset track if the main contributions are not revised.

- The evaluation: The retrieval tests are essential, but more video recognition tests can make the evaluation more solid, e.g., on Egtea, Epic, etc.

- Typo: L246, i.e.Eq.1

---

> ### Author Response · Authors · 2022-08-02
> **Thanks and Response #4 Part B**
>
> > The results look good and impressive, but the readers may wonder how much of them come from the curation? Though without curation, the videos in Ego4D may be not easy to directly use, the performance improvements can be definitely thanks to the data contribution of Ego4D when compared with the first-view data pre-training. Only an "EgoClip" is weird and misleading. Thus, I suggest the authors add the tests to use the original videos of Ego4D in some ways, or if not at least discuss this clearly. If we think about this point, the data contribution of this work and the performance claim may be misaligned.
>
> We provide ablation experiments to study the gains brought from our data curation. **We have three conclusions**:
> 1. The egocentric videos are high-resolution and might be extremely long, which are impossible for end-to-end pretraining. In Sup. A.2, we provide a solution of video compression and chunking that make pretraining possible.
> 2. **Data cleaning** to remove noisy video and narration is meaningful, with 1.5% Acc in Inter and 2.0% Acc in Intra settings of the EgoMCQ development set, and 1.5% mAP and 1.4% nDCG in EPIC-kitchens.
> 3. **Clip pairing strategy** is crucial for untrimmed video without start/end times - our key designs for EgoClip help us reach further gains that 2.7% Acc in Inter and 11.79% Acc in Intra of EgoMCQ, and 2.5% in mAP and 3.2% in nDCG in EPIC.
>
> |         | Data Compression / Chunking | Data Cleaning | Clip Definition       | EgoMCQ  (Inter-video) | EgoMCQ (Intra-video) | Zero-shot EPIC-Kitchens (mAP) | Zero-shot  EPIC-Kitchens (nDCG) |
> | ------- | --------------------------- | ------------- | --------------------- | --------------------- | -------------------- | ----------------------------- | ------------------------------- |
> | Ego4D   | ✓                           | ✗             | Avg length            | 85.13                 | 37.69                | 18.1                          | 10.9                            |
> | Ego4D   | ✓                           | ✓             | Avg length            | 86.66                 | 39.72                | 19.6                          | 12.3                            |
> | EgoClip | ✓                           | ✓             | Our best strategy | **89.36**             | **51.51**            | **22.1**                      | **15.5**                        |
>
> ---
>
> > Considering the method contribution is relatively smaller compared to the data, in my opinion, this work may be more suitable for the benchmark and dataset track if the main contributions are not revised.
>
> It should be emphasized that **our method contributions include**:
> - How to pair clips from untrimmed videos i.e., our contextual variable-length clip pairing strategy Eq.1.
> - How to design a pretraining objective i.e., EgoNCE to meet the properties of Ego4D data.
> - How to seek a reliable development benchmark to quickly iterate the pretraining designs. (e.g, Why EgoMCQ and not video-text retrieval in egocentric domain)
> - How to transfer the one pretrained VLP model to several downstream tasks. (e.g., modify the fine-tuning objective to support both video-text and video-only tasks)
>
> ---
> > The evaluation: The retrieval tests are essential, but more video recognition tests can make the evaluation more solid, e.g., on Egtea, Epic, etc.
>
> In this work, we have already conducted substantial evaluations, including **video-text retrieval** (EPIC-Kitchens), **video-text grounding** (Ego4D NLQ), and also three pure-video tasks: **action recognition** (Charades-Ego, Ego4D OSCC), and **temporal action localization** (Ego4D MQ). It’d be interesting to include more recognition tests. Due to the limited time of the rebuttal phase, we will add more EPIC-Kitchens video recognition experiments in our future revision.
>
> ---
> > Typo: L246, i.e.Eq.1
>
> Thanks for pointing this out, we have fixed it in the revision.

---

> > ### Comment · Reviewer_pWxJ · 2022-08-05
> > **Post-rebuttal**
> >
> > Thanks for the detailed responses. My major concerns are basically addressed, except for the contribution claim and the relation with Ego4D. I suggest a revision on this to make the contribution clear.

---

> > > ### Author Response · Authors · 2022-08-07
> > > **Thanks for Post-rebuttal and Response**
> > >
> > > Thank you and we are glad that your major concerns are basically addressed. Thanks for the suggestion about revising the contribution part and relationships against Ego4D.
> > >
> > > We explicitly recap the contribution of our paper:
> > >
> > > 1. We note that Ego4D has not been proposed for video-language pretraining, and cannot be directly used as its untrimmed videos, no direct video-text pairs, and noisy data. We thus see our clear distinction and contribution in proposing a successful approach to curate a pretraining dataset, our proposed EgoClip. Notably, It is also non-trivial to figure out what is the best way of curating Ego4D to create a pretraining dataset EgoClip, e.g., our pairing approach outperforms the naive strategy with a large margin in the development set, which requires substantial design and experimental validations.
> > >
> > > Along with this distinction from Ego4D, we have a number of contributions in this paper:
> > >
> > > 2. Propose a new development set i.e. EgoMCQ to power rapid design of video-text pretraining i.e. its pretraining dataset and model pretraining objective.
> > >
> > > 3. Design of EgoVLP model to learn rich video-text representations. (Consideration of model architecture/Design of egocentric-friendly pretraining objective)
> > >
> > > 4. Extensive experiments and promising results demonstrate the effectiveness and necessity of EgoVLP. Note that Ego4D has not been used previously for any downstream tasks on other datasets. This is where our work makes significant value.
> > >
> > > The above 4 points constitute a systematic study of Egocentric VLP, where #1 is highly related to Ego4D's contribution and we have explained why it is non-trivial to create EgoClip out of Ego4D.
> > >
> > > Besides, we add a Tab to help you better understand the relationship between each component and video-text pretraining.
> > >
> > > | **Contributions**   | untrimmed data source | model architecture | curated pretraining dataset | development set | pretraining objectives | downstream applications | **Can EgoVLP works?**                                  |
> > > | ------------------- | --------------------- | ------------------ | --------------------------- | --------------- | ---------------------- | ---------------- | ------------------------------------------------------ |
> > > | Ego4D               | ✓                     |                    |                             |                 |                        |                  | **no**, without model                                  |
> > > | +Frozen             | ✓                     | ✓                  |                             |                 |                        |                  | **no**, data cannot directly be used                   |
> > > | +Ours (EgoClip) | ✓                     | ✓                  | ✓                           |                 |                        |                  | **yes**, but hard to validate the pretraining design   |
> > > | +Ours (EgoMCQ)       | ✓                     | ✓                  | ✓                           | ✓               |                        |                  | **yes,** and we want to make better use of data        |
> > > | +Ours (EgoNCE)  | ✓                     | ✓                  | ✓                           | ✓               | ✓                      |                  | **yes,** we need to further validate on downstream tasks |
> > > | +Ours (Transferability studies)  | ✓                     | ✓                  | ✓                           | ✓               | ✓                      | ✓                | **yes, a complete and systematic VLP framework**       |
> > >
> > > ---
> > >
> > >
> > > We hope the above revision could make our contribution clearer. If you have any further suggestions, we'd be very much appreciated.

---

> ### Author Response · Authors · 2022-08-02
> **Thanks and Response #4 Part A**
>
> **Overall:**  *Thank you for your valuable comments about the difference between Ego4D and our work, which help us to better clarify the contribution of our work - a systematic study toward the egocentric video-language pretraining.*
>
> > L184-185: to assure that the scene is not visible during pretraining, we manually remove videos that share multiple views with the videos in EgoClip. Any details about the manually removing?
>
> We first select videos from NUS/Minnesota/Georgia Tech/Indiana sources, which contribute to the multi-camera video data. Then, based on the metadata of the video (i.e. times when videos were collected), we observed that videos collected in the same timeframe tend to be multi-views of the same recording, so we manually group these videos into the same split to ensure the same scene does not appear in another split.
>
> ---
> > **The largest concern**: this work seems not to differentiate the contribution of Ego4D and this submission.
>
> Thanks for raising this point and we will explicitly differentiate ours against Ego4D in the final paper. We’d like to emphasize that based on Ego4D’s existing video data and annotations, it still remains challenging to conduct Egocentric VLP. In this work, we spend significant efforts on **a systematic study of Egocentric VLP** and specifically **address 4 research questions**:
>
> 1. *How to create a video-text pretraining dataset?* (Data selection/Data cleaning/Clip pairing from untrimmed videos).
> 2. *How to develop a pretraining model.* (Consideration of model architecture/Design of ego-friendly pretraining objective)
> 3. *What benchmark we shall evaluate on?* (Create a development set to enable rapid design iteration).
> 4. *Is pre-training necessary in the egocentric domain?* (Transferring strong representation to multiple egocentric benchmarks and validating the necessity and effectiveness of egocentric pertaining.)
>
> We will add this discussion in our final paper.
>
> ---
> > The tab1 is misleading, as the data scale and contribution belong to Ego4D instead of this work. This work conducts mainly the curation and cleaning. So I suggest comparing both Ego4D and EgoClip here to avoid this misleading.
>
> Thanks for the suggestion, we will include the comparison below to Tab. 1 in the final paper. **Note**: Ego4D does not provide direct clips for pertaining (we have proved that clip pairing is crucial in Tab.2) and includes noisy videos and captions. e.g., double-sized stereo video, narration with unsure tag. Thus, *it is not straightforward what is the most suitable pipeline, setting, and strategy for EgoClip creation, especially given its large data scale.*
>
> | Dataset | Ego? | Domain  | Dur (hrs) | #Clips      | #Texts |
> | ------- | ---- | ------- | --------- | ----------- | ------ |
> | Ego4D   | ✓    | diverse | 3.6K      | *untrimmed* | 5.0M   |
> | EgoClip | ✓    | diverse | 2.9K      | 3.8M        | 3.8M   |
>
> ---

---

### Official Review · Reviewer_cPhr · 2022-07-11

**Rating:** 6
**Confidence:** 4
**Soundness:** 3 good
**Presentation:** 3 good
**Contribution:** 3 good

**Summary:**

This paper proposes a method for video+language learning, specifically focused on the ego-centric setting. The paper introduces a new way to mine positive/negative samples for InfoNCE style training, based on sampling by overlapping nouns and verbs in the video caption. The paper introduces a new evaluation setting on the ego4d dataset and evaluates on epic kitchens and charades ego.

**Questions:**

Is there any reason the EGOClip method couldn't be applied to 3rd person videos? None of the approach seems specific to egocentric videos in ways that other videos wouldn't be applicable (e.g., ego-motion).

**Limitations:**

There is some slight discussion on these, but it is very limited. For example, there are many societal impacts of ego-centric videos, such as surveillance, privacy issues, etc. that could be discussed. This goes beyond the impact of the electricity used to train the models.

**Strengths And Weaknesses:**

The proposed mining method for the positive/negative samples is interesting and effective. The results show good performance across multiple datasets and tasks. The state-of-the-art comparisons also show the difference due to different PT datasets and other settings. Overall, the results are fairly complete and show the benefits of the components.

Minor comments:
"Fortunately, with the recent introduction of the massive-scale egocentric video dataset Ego4D [16], it becomes possible to unlock Egocentric VLP."

That is a strong claim, given that in Table 1, Ego4D is still a fraction of the size of other datasets, eg HowTo100M, WebVid, YouTube-Temporal-180M, etc. I wouldn't call Ego4D "massive-scale". The results also are significantly better than the previous results, which suggests this isn't really "unlocking" anything.

There are typos and grammatical errors throughout the paper. It would benefit from careful editing.

Some of the notation is unnecessary complicated. For example, the subscript $i$ in Eq. 1 doesn't add anything. I.e., that section could be revised without the subscript without any loss in clarity. This would also simplify Table 2.

Note that the answer to the licenses is incorrect, Ego4D does not have the MIT license, and has it's own license with certain restrictions.

---

> ### Author Response · Authors · 2022-08-02
> **Thanks and Response #3**
>
> **Overall:** *Thank you for appreciating our work and several great suggestions (e.g., words, EgoClip method, license, social impacts), which help us to improve the paper.*
>
> > Minor comments: "Fortunately, with the recent introduction of the massive-scale egocentric video dataset Ego4D [16], it becomes possible to unlock Egocentric VLP."
> That is a strong claim, given that in Table 1, Ego4D is still a fraction of the size of other datasets, eg HowTo100M, WebVid, YouTube-Temporal-180M, etc. I wouldn't call Ego4D "massive-scale". The results also are significantly better than the previous results, which suggests this isn't really "unlocking" anything.
>
> - **Massive-scale**: The usage of `massive' is referring to the original Ego4D paper. We believe massive is proper since *Ego4D (with 3.6K hours of video) is bigger than all egocentric datasets that have come before it by 2 orders of magnitude*, e.g., the previous largest egocentric dataset EPIC-Kitchens only contains 100 hours. Besides, *egocentric datasets may not be comparable to 3rd person datasets since the data is acquired differently*. Compared with HowTo100M, WebVid, YouTube-Temporal-180M, which automatically scraped a large amount of data from the Web, egocentric videos need to be collected manually using wearable cameras, and the cost is very expensive - they are not common and/or hard to acquire from online sources.
>
> - **Unlocking**: Pre-training on 3rd-person data doesn't necessarily help because of the domain gap [1]. We use unlocking there because previous video-language pretraining mainly focused on the 3rd-person point of view. However, their strong performance cannot transfer to egocentric downstream due to the domain gap. e.g., as illustrated in Tab. 4, with CC3M+WebVid pretraining, the results are still inferior compared to no pretraining baseline JPoSE [2]. We are the first effort to break this barrier.
>
> ---
>
> > There are typos and grammatical errors throughout the paper. It would benefit from careful editing.
>
> Thanks for pointing this out, we have carefully polished the whole paper.
>
> ---
> > Some of the notation is unnecessary complicated. For example, the subscript i in Eq. 1 doesn't add anything. I.e., that section could be revised without the subscript without any loss in clarity. This would also simplify Table 2.
>
> Thanks for the advice, we introduce subscript $i$ to make the formulation more accurate. e.g., variable $\beta_i$ is a varied value determined by the video that sample $i$ belongs to. Also, in Tab.2, subscript makes the variants (c) [$t_{i-1}, t_{i+1}$] easy to understand.
>
> ---
> > Note that the answer to the licenses is incorrect, Ego4D does not have the MIT license and has its own license with certain restrictions.
>
> Thanks for the correction, we have double-checked the Ego4D license and updated it in our revision pdf checklist in red color.
>
> ---
> > Is there any reason the EGOClip method couldn't be applied to 3rd person videos? None of the approaches seems specific to egocentric videos in ways that other videos wouldn't be applicable (e.g., ego-motion).
>
> The key idea of EgoClip creation is contextual variable-length, which is motivated by egocentric videos that record continuous natural events. This should also **help for untrimmed third-person videos**, but *once the videos are edited or inserted by users* - or *the narration is not aligned with the visual content*, such as ASR subtitles - this strategy might not perform well. We will add this discussion in our final paper.
>
> ---
> > There is some slight discussion on these, but it is very limited. For example, there are many societal impacts of ego-centric videos, such as surveillance, privacy issues, etc. that could be discussed. This goes beyond the impact of the electricity used to train the models.
>
> Thanks for the suggestion, we agree that **human privacy** is an important issue closely related to our work. We have included the following in our revision paper in red color.
>
> - *Egocentric VLP learns real-world perception knowledge that may contribute to practical applications such as AI-assisted technologies, robotics, and augmented reality. However, Ego4D videos collected by participants may contain users' privacy and unintended biases, and thus should be used cautiously. We refer the reader to the Ego4D paper about further privacy and societal impacts.*
>
> ---
> **References:**
>
> [1] Li Y, Nagarajan T, Xiong B, et al. Ego-exo: Transferring visual representations from third-person to first-person videos[C] Proceedings of the IEEE/CVF Conference on Computer Vision and Pattern Recognition. 2021: 6943-6953.
>
> [2] Wray M, Larlus D, Csurka G, et al. Fine-grained action retrieval through multiple parts-of-speech embeddings[C] Proceedings of the IEEE/CVF International Conference on Computer Vision. 2019: 450-459.

---

### Official Review · Reviewer_VVdo · 2022-07-12

**Rating:** 4
**Confidence:** 5
**Soundness:** 3 good
**Presentation:** 3 good
**Contribution:** 2 fair

**Summary:**

This paper studies video-language pre-training for egocentric videos.  The authors claim 3 contribution: (1) cleaning the Ego4D dataset to form a large video-text training corpus. (2) proposed EgoNCE to tackle the two challenges (same action in different scenes, and different actions in the same scene) in egocentric video-text pre-training (3) built on top of the Ego4D dataset, the authors introduced EgoMCQ as a development set.

**Questions:**

- Presentation suggestions:
    - Since the paper contains a dataset contribution, it would be great to include a datasheet for it.

**Ethics Review Area:**

["Inadequate Data and Algorithm Evaluation"]

**Limitations:**

The limitations are discussed in Sec 7.

**Strengths And Weaknesses:**

**Strengths**

- The authors studies pre-training for ego-centric videos, which is an important topic for the community, and could be impactful.
- The experimental results are mostly positive, supporting the claims in the paper.

**Weaknesses**

- There is a lack of architectural novelty: the model architecture is essentially the same as Frozen [11].
- L158-165, the authors proposed action-aware positive sampling to address challenge 1 (L151), which relies on the taxonomy annotation specific to the Ego4D dataset and has limited scope, which may not applicable to more general domain ego videos.
- The scene-aware negative sampling is essentially negative sampling from the same video. This technique is often used for tasks like video moment retrieval [Hendricks et. al, Lei et al.], thus it is also not new.
- In the ablations, it is shown that using EgoClip alone is better than using 3rd person view data, either HT100M or CC3M+WebVid2M. It would be interesting to also explore whether the model can benefit from training on a combination of 1st and 3rd view data. E.g., pre-training on EgoClip+CC3M+WebVid2M.
- The two challenges discussed in L150-155 do not seem to unique to 1st person videos. E.g., the same action “running” could happen in a street, a park, or in a playground.
- The claimed contribution (i) (L6) seems trivial, it is almost simply done by doing a bit transform of the Ego4D dataset annotations.
- It is not clear why we need another development set (EgoMCQ), if there are already a bunch of tasks which already has their development set.

Hendricks et. al, Localizing Moments in Video with Natural Language, ICCV 2017

Let et. al, QVHIGHLIGHTS: Detecting Moments and Highlights in Videos via Natural Language Queries, NeurIPS 2021

---

> ### Author Response · Authors · 2022-08-02
> **Thanks and Response #2 Part B**
>
> > The claimed contribution (i) (L6) seems trivial; it is almost simply done by doing a bit transform of the Ego4D dataset annotations.
>
> We respectfully disagree. Given the large data scale and long time needed for validation by training models on top of that, **it is not an easy job to figure out the most suitable transform pipeline, setting, and strategy**. To this end, we have conducted comprehensive explorations including:
> 1. **Data source** (we collect a training set that does not overlap with the five Ego4D benchmarks with 16 tasks, which requires manual checking as multi-view videos exist)
> 2. **Data cleaning** (we carefully filter and process multiple types of noisy video and text, see Sup. A.1)
> 3. **Convert untrimmed videos as clips** (As shown in Tab. 2, we come to a satisfying paring strategy with a large experimental cost. A naive strategy leads to poor performance.)
> 4. **Data chunking** (critical implementation for end-to-end pretraining, see Sup. A.2)
>
> ---
> > It is not clear why we need another development set (EgoMCQ), if there are already a bunch of tasks that already have their development set.
>
> We have stated the need for a development benchmark in L176-181.  We find that **most egocentric benchmarks are domain-specific and focus on video-modality tasks** (see Tab. 1). However, our purpose is to exploit Ego4D’s diversity to learn rich video-text representation. Therefore, it is essential to measure performance on a benchmark highly aligned with the pretraining data domain (Ego4D's diversity) and pretraining objective (video-text alignment). **Existing egocentric downstream benchmarks do not satisfy such characteristics and thus we propose EgoMCQ to satisfy both**.
>
> ---
> > Since the paper contains a dataset contribution, it would be great to include a datasheet for it.
>
> Thanks for the suggestion, we will include an EgoClip datasheet following [1] in our final supplementary material.
>
> [1] Gebru T, Morgenstern J, Vecchione B, et al. Datasheets for datasets[J]. Communications of the ACM, 2021, 64(12): 86-92.

---

> > ### Comment · Reviewer_VVdo · 2022-08-08
> > **Thanks the authors for the response. I would like to keep my original rating.**
> >
> > The authors addressed some of my concerns, but the important ones are still left unclear or not persuasive, especially the contribution of this work over previous dataset Ego4D and previous method frozen. This concern is also shared by Reviewer pWxJ even after response. Overall, I maintain my original rating to reject this paper.

---

> > > ### Author Response · Authors · 2022-08-09
> > > **Thanks for Post-rebuttal and Response**
> > >
> > > Thank you very much for your feedback, we have posted a new response in https://openreview.net/forum?id=nE8_DvxAqAB&noteId=EfkXn2VhnB to recap our contribution (against Ego4D and Frozen model)
> > >
> > > - Our work is motivated by Ego4D but we note that **there is a long way to pave from the Ego4D dataset to Egocentric VLP**, which consists of the **#1 pretraining dataset**, **#2 development set**, **#3 model designs**, and **#4 transferability evaluation**. Among the above 4 points, where *#1 is highly related to Ego4D's contribution and it is non-trivial to create a pretraining dataset EgoClip out of Ego4D*, which requires detailed data curation, substantial design, and experimental validations.
> > >
> > > - We select Frozen as the baseline because its elegant and scalable dual-encoder architecture is representative in state-of-the-art VLP methods. Besides, corresponding to MIL-NCE [1] built on top of the 3rd-person domain's HowTo100M [3], we aim to explore a general pretraining objective i.e., EgoNCE to learn rich video-text representations in 1st-person domains.
> > >
> > > We hope the above claim could make our contribution clearer, and we appreciate any further suggestions.

---

> ### Author Response · Authors · 2022-08-02
> **Thanks and Response #2 Part A**
>
> **Overall**: *Thank you for your comments. First of all, we’d like to emphasize that the position of this paper is to provide a systematic framework for studying egocentric video-language pre-training and to conduct a comprehensive study under such a framework. The framework covers multiple aspects as follows:*
> 1. *How to create a video-text pretraining dataset? (Data selection/Data cleaning/Clip definition from untrimmed videos).*
> 2. *How to design a pretraining model? (Selection of model architecture/Design of pretraining objective)*
> 3. *What benchmark we shall evaluate on? (Create a development set to enable rapid design iteration).*
> 4. *Is pre-training necessary in the egocentric domain? (Extensive transferring experiments to validate the necessity and effectiveness of pretraining.)*
>
> ---
> > There is a lack of architectural novelty: the model architecture is essentially the same as Frozen.
>
> Since **the focus of this paper is not architecture design**, we select Frozen as the basis due to its elegant and scalable dual-encoder designs. *Our conclusions for Egocentric VLP are general and can be transferred to a wide range of architectures*. e.g., Video Swin Transformer. In the future, it’d be interesting to focus on architecture design in another specific paper, where our EgoClip and EgoMCQ could be also useful.
>
> ---
> > L158-165, the authors proposed action-aware positive sampling to address challenge 1 (L151), which relies on the taxonomy annotation specific to the Ego4D dataset and has limited scope, which may not applicable to more general domain ego videos.
>
> 1. First, **the Ego4D dataset is quite general and diverse** – it covers a wide range of 129 daily scenarios, and its taxonomy annotation contains 4336 unique nouns and 1772 unique verbs (115 synonym verbs and 478 synonym nouns). Therefore, its scale and diversity are adequate for covering general egocentric videos. For example, most existing egocentric video datasets (e.g. EPIC-Kitchens, Egtea, UT-Ego, Charades) can be covered by Ego4D taxonomy.
>
> 2. Second, for cases which the taxonomy does not cover, we also can **try other measures e.g., [1] to get relevance for positive sampling**.
>
> ---
> > The scene-aware negative sampling is essentially negative sampling from the same video. This technique is often used for tasks like video moment retrieval [2,3], thus it is also not new.
>
> In contrast to the conventional negative sampling from the same video [2,3], in order to handle the frequent appearance changes in egocentric videos, we specifically design our negative sampling strategy to focus on **temporally adjacent negatives**, which has not been explored in previous approaches. We will add such discussion in the final paper.
>
> ---
> > In the ablations, it is shown that using EgoClip alone is better than using 3rd person view data, either HT100M or CC3M+WebVid2M. It would be interesting to also explore whether the model can benefit from training on a combination of 1st and 3rd view data. E.g., pre-training on EgoClip+CC3M+WebVid2M.
>
> Thanks to this inspiration, we have investigated both 1st and 3rd person view data for pretraining, and interestingly found that combining them together for pretraining causes performance degradation. We leave ideas on **how to combine these two views together as an avenue for future work**.
>
> | PT dataset     | EPIC-Kitchens (1st) | EPIC-Kitchens (1st) | MSR-VTT (3rd) | MSR-VTT (3rd) |
> | -------------- | ------------------- | ------------------- | ------------- | ------------- |
> |                | nDCG                | mAP                 | T2V R@1       | V2T R@1       |
> | EgoClip (1st)  | **23.4**            | **16.5**            | 4.2           | 4.8           |
> | WebVid (3rd)   | 14.2                | 7.8                 | **18.1**      | **15.9**      |
> | EgoClip+WebVid | 22.0                | 15.0                | 17.5          | 15.3          |
>
> ---
> > The two challenges discussed in L150-155 do not seem to unique to 1st person videos. E.g., the same action “running” could happen in a street, a park, or in a playground.
>
> Yes, the same action might happen in 3rd person videos, but it is much more difficult to recognise the action in 1st-person video. This is because
> 1. The person performing action often is behind the camera thus **``running people'' do not appear in the ego video**.
> 2. The ego camera is usually moving and **its view often jitters significantly** or quickly switches between different scenes.
>
> ---
> **References:**
>
> [1] Hendricks et. al, Localizing Moments in Video with Natural Language, ICCV 2017
>
> [2] Let et. al, QVHIGHLIGHTS: Detecting Moments and Highlights in Videos via Natural Language Queries, NeurIPS 2021
>
> [3] Wray, Michael, Hazel Doughty, and Dima Damen. "On semantic similarity in video retrieval." CVPR 2021.

---

### Official Review · Reviewer_93Gf · 2022-07-13

**Rating:** 6
**Confidence:** 3
**Soundness:** 3 good
**Presentation:** 4 excellent
**Contribution:** 3 good

**Summary:**

The paper introduces a first person video-text pretraining dataset called EgoClip comprising 3.8M clip-text pairs chosen from Ego4D. Ego4D is a large-scale dataset comprising first person videos from a large variety of daily human activities. The paper further proposes a pretraining objective called EgoNCE to adapt the video-text contrastive learning to the egocentric domain by mining egocentric aware positive and negative samples. Finally, the paper introduces a new development benchmark called EgoMCQ that is close to the EgoClip dataset and can therefore be used to effectively validate and explore algorithms for EgoClip (which they showcase by exploring EgoNCE on  EgoClip).

EgoClip is sourced from Ego4D. Through a series of criteria, a set of pairs of video and textual narrations are chosen with low amount of noise in terms of video-text misalignment and video-text irrelevance. The final dataset has 2.9K hours of videos with 3.85 million narrations from 129 different scenarios. There are 21.9 clips per minutes with average clip length of 1s and standard deviation of 0.9s. This is followed by designing a contextual variable-length clip pairing strategy where centered around the instantaneous timestamp of narration, a temporal duration of $\beta/\alpha$ is specified such that $\beta$ is the average temporal distance between pairs of consecutive narrations and $\alpha$ is the scale factor computed as the average of all $\beta$s across all videos in EgoClip.

For video-language pretraining, the paper proposes EgoNCE which is an extension of InfoNCE contrastive loss between video-text pairs. EgoNCE adapts InfoNCE for egocentric videos by doing action-aware positive sampling where batch samples that share at least one noun and at least one verb are treated as positive samples. It further has scene-aware negative sampling where different actions in the same scenario are treated as hard negative samples.

Finally, the paper develops a new development benchmark to evaluate on EgoClip called EgoMCQ. The motivation is that standard video-text retrieval may be ineffective since there are substantial duplicate/semantically similar captions in Ego4D. Multiple Choice Question (MCQ) task makes repetitions highly unlikely given a small number of answers. EgoMCQ considers an inter-video setting where five clips originate from different videos, aiming to distinguish instances from different scenarios. EgoMCQ also considers an intra-video setting by grouping five continuous clips together focusing on fine-grained context clues such as hand interaction.

To validate the proposed method, experiments are conducted on five egocentric benchmarks. Experiments show the EgoClip and EgoClip along with EgoNCE perform better than existing methods on all benchmarks. This is because EgoClip helps bring pretraining closer to egocentric downstream tasks. Ablation is conducted on the effect of EgoNCE showing that it is more effective than InfoNCE. Ablation is also conducted to validate the effectiveness of the proposed development benchmark EgoMCQ.


**Questions:**

Questions/Suggestions

1. I would request authors to shed some more light on scenarios where the action-aware positive sampling and scene-aware negative sampling can potential conflict and whether scene-aware negative sampling can introduce false negatives.

2. I would request authors, if possible, to validate the dataset, approach and benchmark on another backbone, such as I3D or Video Swin Transformer, to provide another data point on the ability for the proposed elements to be generalizable across different backbones.

**Limitations:**

I believe the authors have adequately addressed the limitations and potential negative societal impact of their work.

**Strengths And Weaknesses:**

Strengths

1. The proposed components of the work overall have significant novelty and help to improve pretraining for egocentric downstream tasks. EgoClip and accompanying EgoMCQ benchmark will be very helpful to the egocentric community to develop better pretrained models suited for egocentric tasks.
2. The overall paper quality is also high. The paper is easy to read, follow and understand.
3. Extensive evaluation has been performed to evaluate all aspects of the proposed method. The experiments help gather insight on each aspect in a comprehensive manner.

Weaknesses

1. Since egocentric videos are untrimmed and consecutive actions are quite fine-grained, I have some doubts about the scene-aware negative sampling which considers different actions in the same scenario as hard negative samples. But it is possible that different actions in the same scenario can share the same noun. In that case, it appears it might conflict with the action-aware positive sampling. Is it possible that scene-aware negative sampling can introduce false negatives?

2. All experiments for EgoClip seem to be conducted using the Frozen method and TimeSformer backbone. It would be more helpful to see if, just for proof of concept on maybe one setting, how the dataset and approach performs with another backbone.

---

> ### Author Response · Authors · 2022-08-02
> **Thanks and Response #1**
>
> **Overall:** *Thanks for your endorsement of our work and appreciate the full and accurate summary. The comments point out the critical insight of our method, which are thoughtful and inspiring.*
>
> ---
> > I have some doubts about the scene-aware negative sampling which considers different actions in the same scenario as hard negative samples. But it is possible that different actions in the same scenario can share the same noun. In that case, it appears it might conflict with the action-aware positive sampling. Is it possible that scene-aware negative sampling can introduce false negatives?
>
> **Firstly, there is no conflict between the two techniques.** Our definition of action considers both nouns and verbs, with synonyms (L163-164). As you said, scene-aware negative sampling might introduce some negative samples shared with the same noun. e.g., [case1] `open a book` may have a positive [case2] `unfold a book` and a negative [case3] `close a book`. In this way, the model is asked to distinguish their minor differences in terms of verbs, which is make sense.
>
> **Secondly, whether an instance is a False negative or not depends on the semantic granularity.** e.g., [case 2] and [case 3] can both be positive for [case 1] in a coarse-grained setting. But we want the model to recognize the fine-grained differences in the first-person view, thus regard [case3] as negative, a challenging but helpful task.
>
> We acknowledge the limitation of using only one verb and one noun to model action. e.g., `open a book with left hand` and `unfold a book with right hand` are not differentiated as positive samples. How to expand this we leave to future work.
>
> ---
> > All experiments for EgoClip seem to be conducted using the Frozen method and TimeSformer backbone. It would be more helpful to see if, just for proof of concept on maybe one setting, how the dataset and approach performs with another backbone.
>
> Thanks for the advice, we selected Frozen due to its elegant and scalable dual-encoder design. We add an experiment by replacing the video encoder TimeSformer with Video Swin Transformer and train with EgoClip. As stated in Tab, we have three findings: **(i)** Video Swin Transformer also benefits from the EgoClip dataset and reaches significant results in three benchmarks. **(ii)** Video Swin Transformer reaches better performance compared with Timesformer as stronger temporal modeling capacity, which is helpful in egocentric videos. **(iii)** Our EgoNCE brings consistent gains in both backbones, which demonstrates the generality and superiority of our objective design.
>
> | Video backbone         | Pretraining objective | EgoMCQ  (Intra-video) | EgoMCQ  (Inter-video) | EPIC-Kitchens (Zero-shot) | EPIC-Kitchens (Zero-shot) | Charades-Ego (Zero-shot) |
> | ---------------------- | --------------------- | --------------------- | --------------------- | ------------------------- | ------------------------- | ------------------------ |
> |                        |                       | Acc                   | Acc                   | mAP                       | nDCG                      | mAP                      |
> | Timesformer            | InfoNCE               | 89.4                  | 51.5                  | 15.5                      | 22.1                      | 31.2                     |
> | Timesformer            | EgoNCE               | 90.6              | 57.2              | 16.6                  | 23.1                  | 32.1                 |
> | Video Swin Transformer | InfoNCE               | 90.1                  | 53.0                  | 15.8                      | 22.9                      | 31.0                     |
> | Video Swin Transformer | EgoNCE                | **91.7**              | **58.3**              | **16.9**                  | **24.0**                  | **32.5**                 |
> ---

---

### Author Response · Authors · 2022-08-02
**Thanks and Summary of all responses**

We thank all reviewers for their valuable time and constructive feedback. We are grateful for the recognition about **the significance and impact of our work** (R1, R2, R4), **the novelty of our method** (R1, R3), **promising results** (R1, R2, R3, R4), and **clear writing** (R1, R4).

The reviewers raised some concerns, which we have addressed in the below response. Here we *summarize the highlights of our responses*:
- Further conduct experiments using another video backbone e.g., Video Swin Transformer to prove the generality of our proposed components e.g., EgoClip, EgoMCQ. (R1)
- Showing the effect of pretraining with the combination of 1st and 3rd view data. (R2)
- Include a datasheet of our proposed EgoClip dataset. (R2, under preparation)
- Polish paper writing; Correct the license claimed on the paper checklist. (R3)
- Expand the discussion of social impact e.g., user privacy. (R3)
- More discussion to differentiate the contribution of Ego4D and our work. (R4)
- Ablation studies of gains brought by our data curation. (R4)

---

### Author Response · Authors · 2022-08-09
**Thanks AC and all reviewers, Recap our contributions**

> Thanks again to AC and all reviewers for their efforts. We here explicitly rebrief the big picture and contribution of our work (especially against Ego4D) and hope to build a clear understanding for you.

In our work, we study the video-language pretraining in a specific yet significant domain - the 1st-person view, which is motivated by the release of the Ego4D dataset. However, **there is a long way to pave from the Ego4D dataset to Egocentric VLP**, which consists of the pretraining dataset, development set, model designs, and transferability evaluation. *Since they are not as fully explored as their third-person counterparts, thus we need to pioneer them by ourselves - the contribution of our work.*

1. **[pretraining dataset]** We note that Ego4D has not been proposed for video-language pretraining, and cannot be directly used as its untrimmed videos, no direct video-text pairs, and noisy data. We thus see our clear distinction in proposing a successful approach to curate a pretraining dataset, our proposed EgoClip. Notably, It is also non-trivial to figure out what is the best way of curating Ego4D to create a pretraining dataset EgoClip, e.g., our pairing approach outperforms the naive strategy with a large margin in the development set, which requires substantial design and experimental validations.
Along with this distinction from Ego4D, we have a number of contributions in this paper:
2. **[development set]** In the 1st-person domain, there is lacking a satisfactory benchmark that good aligns with pretraining data diversity and focuses on video-text alignment. Therefore, we propose a new development set i.e. EgoMCQ to power rapid design of video-text pretraining i.e. its pretraining dataset and model pretraining objective.
3. **[model designs]** We select Frozen as the baseline because its elegant and scalable dual-encoder architecture is representative in state-of-the-art VLP methods. Besides, corresponding to MIL-NCE [1] built on top of the 3rd-person domain's HowTo100M [2], we aim to explore a general pretraining objective i.e., EgoNCE to learn rich video-text representations in 1st-person domains.
4. **[transferability evaluation]** Extensive experiments and promising results demonstrate the effectiveness and necessity of Egocentric VLP, which will greatly benefit the egocentric community. Note that Ego4D has not been used previously for any downstream tasks on other datasets. This is also where our work makes significant value.

The above 4 points constitute **a systematic study of Egocentric VLP**, where #1 is highly related to Ego4D's contribution and we have explained why it is non-trivial to create EgoClip out of Ego4D.

Furthermore, we add a Table to help you better understand the relationship among each component.

---
| **Contributions**   | untrimmed data source | model architecture | curated pretraining dataset | development set | pretraining objectives | downstream tasks | **Can EgoVLP works?**                                  |
| ------------------- | --------------------- | ------------------ | --------------------------- | --------------- | ---------------------- | ---------------- | ------------------------------------------------------ |
| Ego4D               | ✓                     |                    |                             |                 |                        |                  | **no**, without model                                  |
| +Frozen             | ✓                     | ✓                  |                             |                 |                        |                  | **no**, data cannot directly be used                   |
| +Ours (EgoClip) | ✓                     | ✓                  | ✓                           |                 |                        |                  | **yes**, but unable to validate the pretraining design   |
| +Ours (EgoMCQ)       | ✓                     | ✓                  | ✓                           | ✓               |                        |                  | **yes,** and we want to make better use of data        |
| +Ours (EgoNCE)  | ✓                     | ✓                  | ✓                           | ✓               | ✓                      |                  | **yes,** we need to further validate downstream tasks |
| +Ours (Transferability studies)  | ✓                     | ✓                  | ✓                           | ✓               | ✓                      | ✓                | **yes, a complete and systematic VLP framework**       |
---

**Reference**

[1] Miech A, Alayrac J B, Smaira L, et al. End-to-end learning of visual representations from uncurated instructional videos[C] ICCV. 2020: 9879-9889.

[2] Miech A, Zhukov D, Alayrac J B, et al. Howto100m: Learning a text-video embedding by watching hundred million narrated video clips[C] ICCV. 2019: 2630-2640.

---

### Meta-Review · Area_Chair_ZqBV · 2022-08-24

**Recommendation:** Accept
**Confidence:** Less certain

**Metareview:**

For egocentric video-language pretraining, this paper creates a 1st-person video-text pretraining dataset, proposes a new contrastive loss EgoNCE, and builds a new benchmark EgoMCQ.  Although the contribution of this work is somewhat incremental, its motivation, experimentation and organization are good. Besides, all reviewers agree that egocentric video-language pretraining is an important topic for the community. I hence suggest accepting it.



**Award:**

No

---

### Decision · Program_Chairs · 2022-09-14

Accept